# Numerical Simulation of the Flow in Two-Phase Supersonic Underexpanded Gas–Particle Jets Exhausting into a Slotted Submerged Space

Sergey Kiselev *, Vladimir Kiselev and Viktor Zaikovskii

Laboratory of Physics Multiphase Media, Khristianovich Institute of Theoretical and Applied Mechanics Siberian Branch of the Russian Academy of Science, 630090 Novosibirsk, Russia
* Correspondence: kiselev@itam.nsc.ru; Tel.: +7-913-932-74-79

**Abstract:** A simplified 2D model for calculating two-phase gas–particle flows in a slot space has been developed. The model can be used for fast calculation and estimation of supersonic-flow parameters in the slot space. Using this model, a numerical simulation of the flow in two-phase gas–particle supersonic jets exhausting into a submerged slot space bounded by two parallel disks was performed. The presence of particles led to the splitting of the gas jet into an internal two-phase jet and an external gas jet. In the present study, we investigated the structure of a two-phase jet as dependent on the spacing between the disks for conditions of cold spraying. A new effect was found in the flow at a small spacing between the disks (of the order of 0.2 mm) and a high-velocity internal two-phase gas–particle jet was formed. The distribution of the concentration of particles in the particle jet proved to be essentially non-uniform, with a caustic formed at the upper jet boundary.

**Keywords:** gas–particle jets; slotted submerged space; supersonic flow; numerical simulation

## 1. Introduction

The possibility of using radial nozzles for spraying coatings onto inner pipe surfaces was demonstrated in [1]. The radial nozzle is formed by two parallel disks, in between which the gas flow undergoes acceleration together with microparticles. When microparticles collide with the pipe surface, they adhere to it providing that their velocity is high enough. In [1,2], the influence of the inter-disk spacing on the structure of the gas flow between the disks was examined. In [1,2], the motion of particles was calculated for their low volume concentration $c < 0.1\%$ at which the influence of particles on the gas flow could be ignored. On increasing the particle volume concentration $c$ in excess of 0.1%, the effect of particles on the gas flow can no longer be neglected.

Mathematical models used to describe two-phase gas–particle mixtures can be divided into two classes. The first class of mathematical models is used to describe flows in which the volume concentration of particles $c$ is high, $c > 5\%$ [3]. When calculating such two-phase flows, it is necessary to take into account collisions between particles [3–5]. This is achieved by introducing the collision integral into the kinetic equation for particles [3,4]. In addition, when calculating the drag force acting on a particle, the constraint phenomena become significant [6]. If the distance between particles is equal to several particle diameters, then it is necessary to take into account the pseudo-turbulence due to velocity fluctuations induced by surrounding particles [7,8].

The second class of mathematical models refers to the case of a low volume concentration of particles, $c < 5\%$. In this case, collisions between particles, as well as the effects due to constraint and pseudo-turbulence, can be neglected. For calculating two-phase flows, a continuum-discrete model is used, in which the gas flow is described using the gas-dynamics equations with source terms in the right-hand sides, and the particles are

described using the collisionless kinetic equation [9–11]. The collisionless kinetic equation for particles is solved by the method of Lagrangian particles moving relative to the Eulerian coordinate system [10,11]. Note that the idea of the Lagrangian method for calculating particles was formulated earlier in [12,13]. The method of Lagrangian calculation of particles was used in [14,15] to calculate the interaction of a shock wave with a cloud of particles. In [16–18], the Euler-Lagrangian method was used to calculate the acceleration of microparticles and their focus on Laval micronozzles.

In the present paper, we study the flow of two-phase gas–particle jets in the case of $0.1\% < c < 5\%$. Two-phase gas flows with particles exhausting from a rectangular channel into a slot space are considered. In this case, under the action of the supersonic gas flow the particles are accelerated in the narrow slot space. As a result, a narrow jet of gas–particle mixture is formed at the exit from the slot space. Particles move at a speed of several hundred meters per second. When particles impinge onto an obstacle, a strong coating is formed on it. In the literature, this method of coating application is known as the Cold Spray method [1]. As noted above, the use of round discs makes it possible to adapt this method for applying coatings onto the inner surface of pipes [1]. For calculating the supersonic jet flow, on the basis of the continuum–discrete model of [9,10], a numerical two-dimensional (2D) model has been developed.

## 2. Problem Statement

For refining the 2D model, experiments and numerical calculations were carried out in which the air flow in a slot space without particles was studied (see Figure 1). Figure 1a,b show a diagram of the experimental setup, consisting of parallel disks 1 and 2 mounted on a rod 3. The disks created a slot space with a width of $h = 0.2$ mm. In between the disks, a spacer (shaded area in Figure 1b) was located, in which a cut of length $a$ and transverse size $b$ was made (see Figure 1b). As a result, a channel occupying the region $r_1 < x < r_2$, $-b/2 < y < b/2$, $-h/2 < z < h/2$ with dimensions $r_0 = 5$ mm, $r_1 = 9$ mm, $r_2 = 20$ mm, $b = 4$ mm and $x_i = 6$ mm was formed. The slot space is bounded by a circle of radius $r_e = 36$ mm. From prechamber 4, gas under pressure $p_0 = 1$ MPa and temperature $T_0 = 291$ K was supplied into the channel, from which it exhausted into the slot space $r_2 < x < r_e$. On the outer disk, a number of pressure holes with a diameter of $d_i = 0.8$ mm were made, using which the static pressure was measured in the steady flow with 0.5% accuracy. A soot-oil coating was applied onto the surface of the outer disk in order to make it possible to identify the boundaries of the jet in the slot space.

From Figure 1a,b, it follows that the gas flow at the channel inlet will be three-dimensional; that is why, strictly speaking, its modeling must be performed in 3D formulation. As numerical calculations show, the gas velocity in the slot space is of the order of $u \approx 10^3$ m/s; therefore, for $h = 0.2$ mm, we obtain a Reynolds-number value Re $\approx 10^4$, at which the gas flow will be turbulent; therefore, in numerical calculations it is necessary to use the LES (Large Eddy Simulation) approach [19,20]. In such cases, for performing 3D calculations, high-speed multiprocessor computers with a large amount of memory are required. In [19], the numerical simulation of supersonic flow in a Laval nozzle with liquid-jet injection was performed using 44 million cells. The use of so many cells creates serious difficulties in numerical simulations. To overcome these difficulties, in the present paper, we propose an approximate 2D calculation model. This model makes it possible to carry out fast calculations of supersonic gas–particle flows in the slot space between disks. The physical basis for this model is the condition that the width of the slot space is small compared to its length, $h/r_e \approx 0.0056 << 1$. The inlet region is separated out from the slot space by a narrow channel with a subsonic gas flow. In this case, the 3D disturbances arising at the channel inlet will be damped in the channel due to viscosity. The performance of the proposed 2D model is substantiated by a satisfactory agreement between calculated and experimental data.

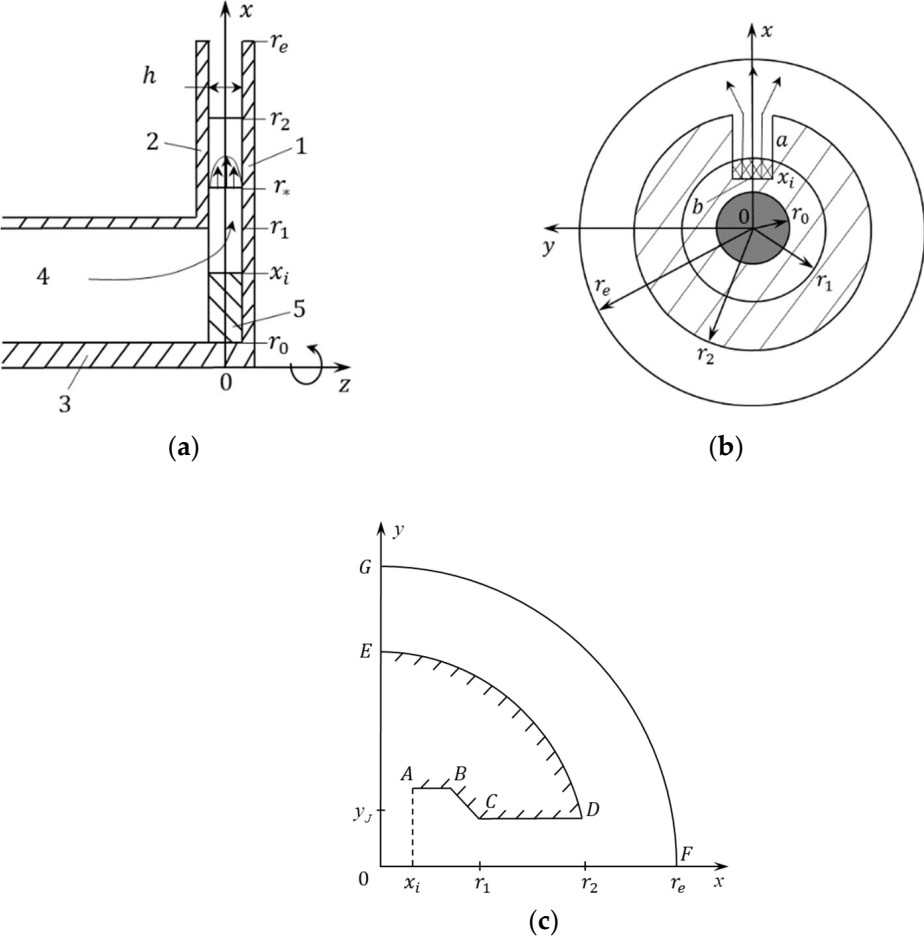

**Figure 1.** The flow region in the experiment: (**a**) in the $(x, z)$-plane; (**b**) in the $(x, y)$-plane; (**c**) computational domain in the $(x, y)$-plane.

### 2.1. Calculation Procedure

Consider the problem regarding a gas–particle flow in a region formed by a plane channel and an adjacent region of a slot space formed by two parallel disks (Figure 1a,b). The upper half of the computational domain is shown in Figure 1c. The lower half of the flow is located symmetrically about the abscissa axis.

The $x$ and $y$ axes are located in the plane parallel to the planes of the channel and disks, at $z = 0$. The $z$-axis is directed normally to the $x$, $y$-plane, and the $x$-axis coincides with the axis of symmetry (see Figure 1b). The distances between the sidewalls of the channel and the disks are identical, $\Delta z = h$. The flow region consists of two sub-regions (see Figure 1). The first sub-region is the channel bounded by the contour $ABCD$, and the second sub-region is the slot space in between the disks, bounded by the contour $EFDG$.

### 2.2. Equations for Gas Flow in the Channel Approximation

The gas–particle flow is described by flow equations in the channel approximation. In this approximation, the equations for the gas are obtained by averaging the gas quantities over the transverse coordinate $-h/2 < z < h/2$, a procedure that makes them dependent only on the coordinates $x$, $y$. For obtaining equations in the channel approximation, we write equations for a viscous heat-conducting gas in divergent form:

$$\frac{\partial \rho}{\partial t} + \frac{\partial \rho v_i}{\partial x_i} = 0, \ \frac{\partial \rho v_i}{\partial t} + \frac{\partial}{\partial x_j}\left(\rho v_i v_j + p\delta_{ij}\right) = \frac{\partial \tau_{ij}}{\partial x_j}, \ \tau_{ij} = 2\mu_t\left(S_{ij} - \frac{1}{3}S_{kk}\delta_{ij}\right),$$

$$\frac{\partial}{\partial t}\left(\rho\left(E + \frac{v^2}{2}\right)\right) + \frac{\partial}{\partial x_j}\left(\rho v_j\left(H + \frac{v^2}{2}\right)\right) = \frac{\partial(\tau_{ij}v_i)}{\partial x_j} - \frac{\partial q_j}{\partial x_j}, \ S_{ij} = \frac{1}{2}\left(\frac{\partial v_i}{\partial x_j} + \frac{\partial v_j}{\partial x_i}\right), \quad (1)$$

$$p = (\gamma - 1)\rho E, \ E = C_V T, \ H = E + \frac{p}{\rho}, \ q_j = -\lambda_t\frac{\partial T}{\partial x_j},$$

where the subscripts $i$, $j$, $k$ denote the components $x$, $y$, $z$, and the summation is performed over repeated indices. On the disk surface, $z = \pm h/2$, system (1) must obey the no-slip boundary conditions $v_x = v_y = v_z = 0$ and the condition of zero flux of heat into the walls (adiabaticity condition $q_z = 0$). In the case of a narrow channel (see Figure 1a) in Equation (1), one can put

$$v_z = 0, \ \partial p/\partial z = 0 \quad (2)$$

Taking into account conditions (2) and inequalities $\tau_{zx} >> \tau_{xx}$, $\tau_{zx} >> \tau_{yx}$, $\frac{\partial \tau_{zx}}{\partial z} >> \frac{\partial \tau_{zx}}{\partial x}$, $\frac{\partial \tau_{zx}}{\partial z} >> \frac{\partial \tau_{zx}}{\partial y}$, $q_z >> q_x$, $q_z >> q_y$, we rewrite Equation (1) as:

$$\frac{\partial \rho}{\partial t} + \frac{\partial \rho v_x}{\partial x} + \frac{\partial \rho v_y}{\partial y} = 0, \ \frac{\partial \rho v_x}{\partial t} + \frac{\partial}{\partial x}\left(\rho v_x^2 + p\right) + \frac{\partial}{\partial y}\left(\rho v_x v_y\right) = \frac{\partial \tau_{zx}}{\partial x},$$

$$\frac{\partial \rho v_y}{\partial t} + \frac{\partial}{\partial x}\left(\rho v_x v_y\right) + \frac{\partial}{\partial y}\left(\rho v_y^2 + p\right) = \frac{\partial \tau_{zy}}{\partial y}, \ q^2 = v_x^2 + v_y^2,$$

$$\frac{\partial}{\partial t}\left(\rho\left(E + \frac{q^2}{2}\right)\right) + \frac{\partial}{\partial x}\left(\rho v_x\left(H + \frac{q^2}{2}\right)\right) + \frac{\partial}{\partial y}\left(\rho v_y\left(H + \frac{q^2}{2}\right)\right) = \frac{\partial(\tau_{xz}v_x)}{\partial z} + \frac{\partial(\tau_{yz}v_y)}{\partial z} - \frac{\partial q_z}{\partial z}$$

Integrating these equations across the slot region and taking into account the boundary conditions at $z = \pm h/2$, we obtain:

$$\frac{\partial \rho}{\partial t} + \frac{\partial \rho u}{\partial x} + \frac{\partial \rho v}{\partial y} = 0, \ \frac{\partial \rho u}{\partial t} + \frac{\partial}{\partial x}\left(\rho u^2 + p\right) + \frac{\partial}{\partial y}\left(\rho uv\right) = -\tau_x,$$

$$\frac{\partial \rho v}{\partial t} + \frac{\partial}{\partial x}\left(\rho uv\right) + \frac{\partial}{\partial y}\left(\rho v^2 + p\right) = -\tau_y, \ \tau_x = 2\tau_{zx}|_{z=h/2}, \ \tau_y = 2\tau_{zy}|_{z=h/2},$$

$$u = \frac{1}{h}\int_{-h/2}^{h/2} v_x dz, \ \frac{\partial}{\partial t}\left(\rho\left(E + \frac{q^2}{2}\right)\right) + \frac{\partial}{\partial x}\left(\rho u\left(H + \frac{q^2}{2}\right)\right) + \frac{\partial}{\partial y}\left(\rho v\left(H + \frac{q^2}{2}\right)\right) = 0, \quad (3)$$

$$v = \frac{1}{h}\int_{-h/2}^{h/2} v_y dz.$$

Note that when the gas flows from the prechamber into the channel, the streamlines become bent (see Figure 1a); therefore, Görtler vortices can appear in the boundary layer of the channel, which are then carried into the slot space. In this case, in averaging Equation (1) over the transverse coordinate, instead of conditions (2), conditions $\overline{v}_z = 0$, $\partial \overline{p}/\partial z = 0$ must be used, where $\overline{v}_z = \int_{-h/2}^{h/2} v_z dz/h$ and $\overline{p} = \int_{-h/2}^{h/2} p dz/h$. After averaging Equation (1), we again obtain Equation (4), with the only difference being that the energy equation will include, instead of $q^2$, the sum $q^2 + \overline{v_z^2}$. In the case of supersonic flows, the condition $q^2 >> \overline{v_z^2}$ is satisfied in the slot space; that is why Equation (3) can also be used in the presence of Görtler vortices in the boundary layer.

For closing the system (3), it is necessary to determine the force $\tau_x$, $\tau_y$ that acts on the gas from the side of disk walls. By analogy with the turbulent flow in a pipe [21], we put

$$\tau_x = C_W \rho u q/h, \ \tau_y = C_W \rho v q/h, \ C_W = \frac{12}{\mathrm{Re}} + \frac{0.06}{(2\mathrm{Re})^{0.25}}, \ \mathrm{Re} = \frac{\rho q h}{\mu}. \quad (4)$$

The drag coefficient $C_W$ involves two terms. The first term corresponds to the laminar and the second to the turbulent flow. In the literature, the second term is called the Blasius formula [21]. The numerical values in the formula for $C_W$ were chosen from the condition of matching the profile $p(x)$ in the channel obtained in the calculation by Formula (3) and in the experiment to the profile $p(x)$ in the slot space. Formula (4) is valid in the case in which there occurs a closure of the boundary layers that appear on the outer and inner

disks. Let us estimate the distance from the channel inlet to the point of closure of the boundary layers $\Delta x_* = r_* - r_1$ (see Figure 1a). The thickness of the turbulent layer on a plate increases with the coordinate $x$ according to the linear law $\delta = v_* x / u$, where $v_*$ is the characteristic velocity of turbulent fluctuations [22]. Using the expressions for the frictional stress on a plate $\sigma_{xz} = \rho v_*^2 = C_f \rho u^2 / 2$, we obtain $u / v_* = \sqrt{2/C_f}$ [22]. In the case of turbulent flow, we have $C_f \approx 0.06 / \mathrm{Re}^{1/4}$ and, therefore, at $\mathrm{Re} \approx 10^4$ we have $u / v_* \approx 20$. Substituting the values $\delta = h/2$, $x = \Delta x_*$, and $u / v_* \approx 20$ into the formula $\delta = v_* x / u$, we obtain: $\Delta x_* \approx 10h$. On the assumption that $h = 0.2$ mm, we find the estimate $\Delta x_* \approx 2$ mm. It follows from here that Formula (4) is valid for the gas flow in the slot space.

*2.3. Equations for Gas and Particles in the Channel Approximation*

When particles are present in the gas flow, it is necessary to add to the right-hand side of the motion Equation (3) the force $F_x$, $F_y$ that acts on the gas from the side of particles. To the right side of energy Equation (3), it is necessary to add the heat flux $Q_e$ due to the exchange of heat between the gas and particles and due to the heating of the gas by the friction force acting between the gas and particles. The motion of particles is described by the collisionless kinetic equation for the particle distribution function $f$ [10,11]. The gas flow is described by the system of Equation (3), in the right parts of which the force of gas interaction with particles and the heat flux from gas to particles are added. The complete system of gas–particle equations has the form:

$$\frac{\partial g}{\partial t} + \frac{\partial F}{\partial x} + \frac{\partial G}{\partial y} + H = 0,$$

$$g = \begin{pmatrix} \rho \\ \rho u \\ \rho v \\ \rho\left(e + \frac{\mathbf{v}^2}{2}\right) \end{pmatrix}, F = \begin{pmatrix} \rho u \\ \rho u^2 + p \\ \rho u v \\ \rho u\left(e + \frac{p}{\rho} + \frac{\mathbf{v}^2}{2}\right) \end{pmatrix}, G = \begin{pmatrix} \rho v \\ \rho v u \\ \rho v^2 + p \\ \rho v\left(e + \frac{p}{\rho} + \frac{\mathbf{v}^2}{2}\right) \end{pmatrix},$$

$$H = \begin{pmatrix} 0 \\ \tau_x - F_x \\ \tau_y - F_y \\ -uF_x - vF_y - Q_e \end{pmatrix}, \ E = C_V T, \ p = (\gamma - 1)\rho E, \ q = \sqrt{u^2 + v^2},$$

$$F_x = -\int m_p a_{px} f \, du_p dv_p dT_p, \ F_y = -\int m_p a_{py} f \, du_p dv_p dT_p, \ \mathrm{M} = \frac{q}{a},$$

$$Q_e = \int m_p \left((u - u_p)a_{px} + (v - v_p)a_{py} - C_s q_p\right) f \, du_p dv_p dT_p, a_{px} = \frac{u - u_p}{\tau_p}, \ a_{py} = \frac{v - v_p}{\tau_p},$$

$$q_p = \frac{T - T_p}{\tau_T}, \mu = \mu_0 \frac{T_0 - T_c}{T - T_c}\left(\frac{T}{T_0}\right)^{3/2}, \ \lambda = \frac{C_p \mu}{\mathrm{Pr}}, \ \frac{1}{\tau_p} = \frac{3}{4}\frac{\rho q_{12}}{\rho_p d_p}C_d(\mathrm{Re}_{12}, \ \mathrm{M}_{12}),$$

$$\frac{1}{\tau_T} = \frac{6\lambda \mathrm{Nu}}{c_s \rho_p d_p{}^2}, \mathrm{Nu} = 2 + 0{,}6\mathrm{Re}^{1/2}\mathrm{Pr}^{0,33}, \ q_{12} = \sqrt{(u - u_p)^2 + (v - v_p)^2},$$

$$\mathrm{Re}_{12} = \frac{\rho q_{12} d_p}{\mu}, \ \mathrm{M}_{12} = \frac{q_{12}}{a}, \ a = \sqrt{\gamma \frac{p}{\rho}},$$

$$\frac{\partial f}{\partial t} + u_p \frac{\partial f}{\partial x_p} + v_p \frac{\partial f}{\partial y_p} + \frac{\partial a_{px} f}{\partial u_p} + \frac{\partial a_{py} f}{\partial v_p} + \frac{\partial q_p f}{\partial T_p} = 0, f = f(t, x_p, y_p, u_p, v_p, T_p),$$

$$n = \int f \, du_p dv_p dT_p$$

(5)

The $f$ is the distribution function of particles in the phase space $t$, $x_p$, $y_p$, $u_p$, $v_p$, $T_p$. The particle drag coefficient $C_d$ depends on the relative Reynolds and Mach numbers $\mathrm{Re}_{12}$, $\mathrm{M}_{12}$ [23]. The $F_x$, $F_y$, $Q_e$ are the components of the force acting from the side of particles and the flux of energy from particles to the gas flow [9–11].

*2.4. Method for Calculating Gas–Particle Flows*

In this paper, two problems are considered. The first problem is that the calculation of the gas flow, without particles (here, Equations (3) and (4) are to be solved), is merely methodical. The second problem concerns the calculation of the flow of the gas–particle mixture (here, Equations (4) and (5) are to be solved), which is a new problem of scientific interest. Both problems were solved numerically by the relaxation method in the region comprising a rectangular channel and a slotted flooded slot space (see Figure 1c). At the initial time, in the middle of the channel a discontinuity of gas parameters was set, to the

left of which were parameters $p_0$, $T_0$ like those in the high-pressure chamber, and to the right, parameters $p_a$, $T_a$ like those in normal atmosphere were specified. In the second region of the flow, parameters at normal atmospheric pressure and temperature, $p_a$, $T_a$, were set. The gas was discharged into the atmosphere under normal conditions; therefore, the corresponding conditions were set at the computational-domain boundary *FG*. For supersonic flow, M $= q/a > 1$, symmetry conditions on the boundary *FG* were adopted, and for subsonic flow, M $< 1$, relations on the Riemannian characteristic were used, where M is the Mach number of the gas flow. At the channel inlet *AD*, parameters $p_0$, $T_0$, like those in the high-pressure chamber, were adopted, and the gas velocities were determined from the relation on the characteristic. At the channel inlet *AD*, over the section $y < y_J$ the particle flux $J_p$ was set. The impermeability condition was set at the boundaries *AB*, *BC*, and the condition of symmetry was set on the sections *DE*, *EF*, *CG*. For particles on the axis of symmetry *DE*, *EF* and on the boundary *AB*, the mirror reflection condition was adopted.

The equations for the gas in (3) and (5) were solved on a curvilinear Eulerian grid using the modified explicit Lax–Wendroff scheme, whose order of accuracy was reduced to unity in order to suppress the oscillations at discontinuities [2]. The modified Lax–Wendroff difference scheme used is described in Appendix A. The collisionless kinetic Equation in (5) was solved in Lagrangian variables by the particle cell method [10,11]. The particles that were introduced into the computational domain were divided into cells in such a way that each cell contained particles of the same diameter with the same velocity and temperature. In this case, during the motion of particles, their number in each cell was preserved and the equations of motion of the cells coincided with the characteristics of the kinetic Equation in (5):

$$\frac{dx_p}{dt} = u_p, \quad \frac{dy_p}{dt} = v_p, \quad \frac{du_p}{dt} = a_{px}, \quad \frac{dv_p}{dt} = a_{py}, \quad \frac{dT_p}{dt} = q_p \tag{6}$$

For calculating the interaction force $a_{px}$, $a_{py}$ and the flux of heat $q_p$ between the gas and particles, the gas parameters were projected onto the particle cells. The gas parameters in the $i$-th particle cell were found by the linear interpolation method using the formula

$$g_i^n = \sum_{j=1}^{J} S_{ij}^n g_j^n \Big/ \sum_{j=1}^{J} S_{ij}^n \tag{7}$$

where the summation over $J$ is carried out over the gas cells with which the $i$-th particle cell intersects (see Figure 2a), and the quantity $g$ assumes the value of one of the gas parameters—density $\rho$, velocity $u$, $v$, or temperature $T$.

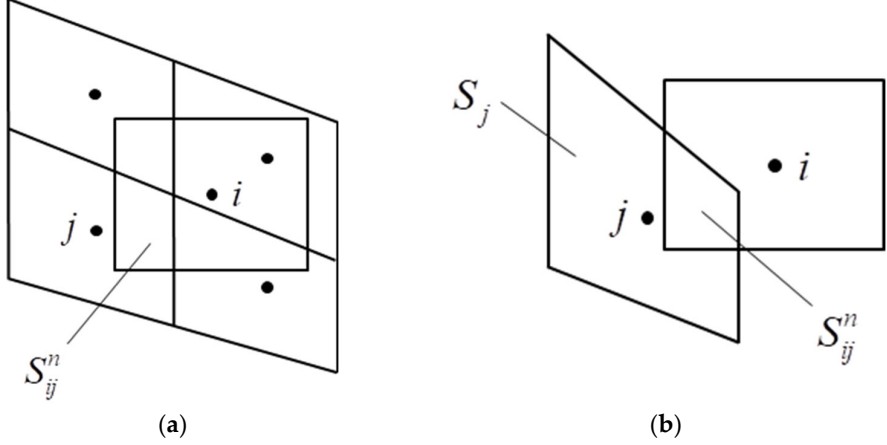

(**a**)  (**b**)

**Figure 2.** Diagram illustrating the determination of gas characteristics in a particle cell grid: (**a**) the gas parameters in the $i$-th particle cell; (**b**) the particle parameters in the $j$-th Eulerian gas cell.

The concentration of particles $n_j^n$, the interaction forces $F_{xj}^n$, $F_{yj}^n$, and the flux of energy $Q_{ej}^n$ in the $j$-th Eulerian gas cell (see Figure 2b) were calculated by the formulas

$$n_j^n = \sum_{i=1}^{I} \frac{S_{ij}^n n_i}{S_j}, \quad F_{xj}^n = -\sum_{i=1}^{I} m_{pi} a_{pxi}^n \frac{S_{ij}^n n_i}{S_j}, \quad F_{yj}^n = -\sum_{i=1}^{I} m_{pi} a_{pyi}^n \frac{S_{ij}^n n_i}{S_j},$$

$$Q_{ej}^n = \sum_{i=1}^{I} m_{pi} \left( \left( u_i^n - u_{pi}^n \right) a_{pxi}^n + \left( v_i^n - v_{pi}^n \right) a_{pyi}^n - c_{si} q_{pi}^n \right) \frac{S_{ij}^n n_i}{S_j}, \quad (8)$$

where the summation in Formula (8) is performed over all $i$ particle cells intersecting with the $j$-th gas cell.

*2.5. Calculated Parameters of the Problem*

As noted above, this paper considers two problems. In the first problem, the gas flow in the slot space is studied, and in the second problem, we examine the flow of a gas–particle mixture in the slot space. In both problems, one and the same computational domain is considered, involving a channel and a slot space formed by two disks (see Figure 1).

As noted above, the computational domain consisted of a channel and a disk (see Figure 1). The rectangular channel consisted of inlet and outlet parts connected together by an inclined wall. The height of the inlet part of the channel is $y_A = 4$ mm, and the height of the outlet part, $y_B = 2$ mm. The length of the inlet part of the channel was $\Delta x_{AB} = 2$ mm, and that of the inclined and outlet parts, respectively, $\Delta x_{BC} = 4$ mm and $\Delta x_{CD} = 12$ mm. We studied the flow of a two-phase jet in between closely and widely spaced disks. The inner radius of both disks had identical values equal to $r_1 = 20$ mm. The outer radius of the narrow disks was $r_2 = 36$ mm, and the radius of the widely spaced disks, $r_2 = 90$ mm. In the present study, we investigated the influence of the width of the slot space on the flow of the two-phase jet in between the disks. For this reason, calculations were carried out for different widths $h$ of the slot space.

As the working gas, air with the following parameters was chosen: $C_V = 732$ J/(kg $\cdot$ K), $\gamma = 1.4$, $\mu_0 = 1.827 \cdot 10^{-5}$ kg/(m $\cdot$ s), $T_0 = 291$ K, $T_1 = 120$ K, and Pr $= 0.72$. We analyzed the motion of aluminum particles with the following parameters: $C_s = 880$ J/(kg $\cdot$ K), $\rho_p = 2700$ kg/m$^3$, particle diameter $d_p = 10$ μm. The pressure and temperature of the gas under normal atmospheric conditions were $p_a = 0.1$ MPa and $T_a = 283$ K. The gas pressure and temperature at the inlet to the channel were $p_0 = 1$ MPa and $T_0 = 283$ K. Figure 3 shows the difference grid on which our numerical calculations were carried out. The total number of cells in the computational domain was $N_c = 51,783$. The number of calculated grid nodes in the plane channel was $N_k = 174 \times 22 = 3828$, and that in the space between the disks, $N_d = 139 \times 345 = 47,955$.

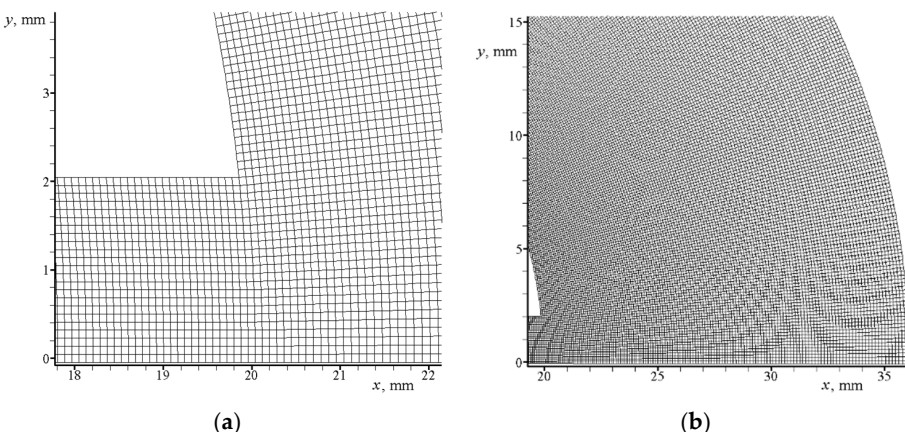

**(a)**　　　　　　　　　　　　　　　　　　**(b)**

**Figure 3.** Difference grid: (**a**) in the vicinity of the channel outlet into the slot space; (**b**) in the slot space.

Particles were introduced by specifying on the boundary $AD$ a row of cells parallel to the $y$-axis with a step $\Delta y_c$. The total number of such cells was $N_c = 30$. The sizes of the particle cells, $\Delta x_c = 0.08$ mm and $\Delta y_c = 0.08$ mm, were chosen such that their value would be of the order of the size of the grid cells for the gas. The volumetric concentration of particles in the cells was $c = 9 \cdot 10^{-3}$ ($c = nV_p$, where $V_p$ is the particle volume). The initial velocities and temperatures of the particles at the boundary $AD$ were $u_p = 50$ m/s and $T_p = 283$ K. After some row of particle cells was displaced to a distance $\Delta x_c$ inside the channel, a new row of cells was introduced. The rate of the particle flow was calculated by the formula $j_s = \rho_p c u_p y_J h$. The number of particle cells in the computational domain was $N_s = 4146$.

## 3. Results and Discussion

### 3.1. Calculation of the Gas Flow without Particles

To check the efficiency of the proposed 2D calculation Model (3), we carried out a numerical simulation of the gas flow without particles between two disks located at a distance $h = 0.2$ mm. The radius of the outer disk was $r_e = 36$ mm, and the number of cells, $N_c = 51,783$. Suppose that at the initial time $t = 0$, in the channel filled with the gas at rest (air) with temperature $T_0 = 283$ K, there exists a pressure jump: $p_0 = 1$ MPa at $x < 12.5$ mm, and $p_0 = 0.1$ MPa at $x > 12.5$ mm (line 0 in Figure 4a). As a result of the disintegration of the discontinuity, there arises an unsteady flow, which is established with the passage of time (see Figure 4). In the present paper this flow is assumed steady, providing that the amplitude of flow-rate fluctuations at the exit from the slot space satisfies the inequality $\Delta j_g / j_g \leq 2\%$.

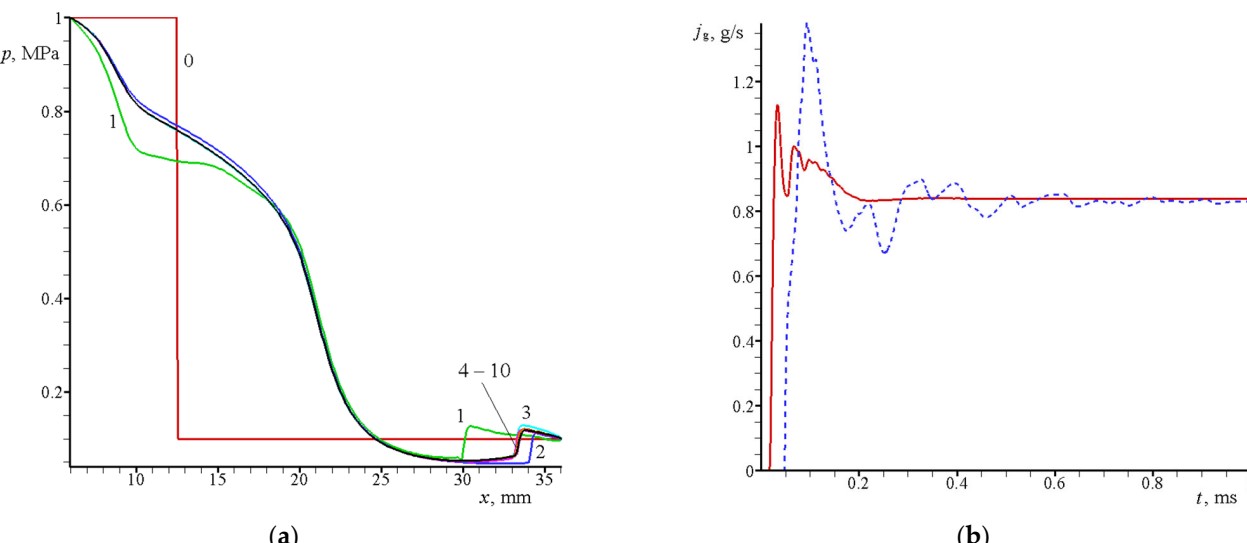

(**a**)                                    (**b**)

**Figure 4.** Calculation results for the unsteady gas flow exhausting into the slot space of height $h = 0.2$ mm: (**a**)—dependence of gas pressure on the axis $y = 0$ at ten times $t_i$ following at time intervals $\Delta t = 0.1$ ms; (**b**)—dependence of gas-flow rate on time in the inlet and outlet sections of the channel (solid red line and blue dashed line, respectively).

The steady flow is illustrated in Figure 5. It is seen that there is a subsonic flow in the channel. When the stream leaves the channel and enters the slot space, it turns around and accelerates to a supersonic speed in a centered rarefaction wave.

The supersonic flow in the slot space is limited on the right by the shock wave (see Figure 5b). In the case when the deceleration force acting from the side of the disks is neglected ($\tau_x = 0$, $\tau_y = 0$), the flow in the channel and in the slot space proves to be supersonic (see the dashed line in Figure 5a).

To check the convergence of our numerical calculations, such calculations were also performed for the numbers of cells $N_c = 51,783$, $N_c' = 207,132$, and $N_c'' = 828,528$. Those

calculations were found to yield close results (see Figure 6) and, therefore, all subsequent calculations (unless stated otherwise) were carried out on a grid involving 51,783 cells.

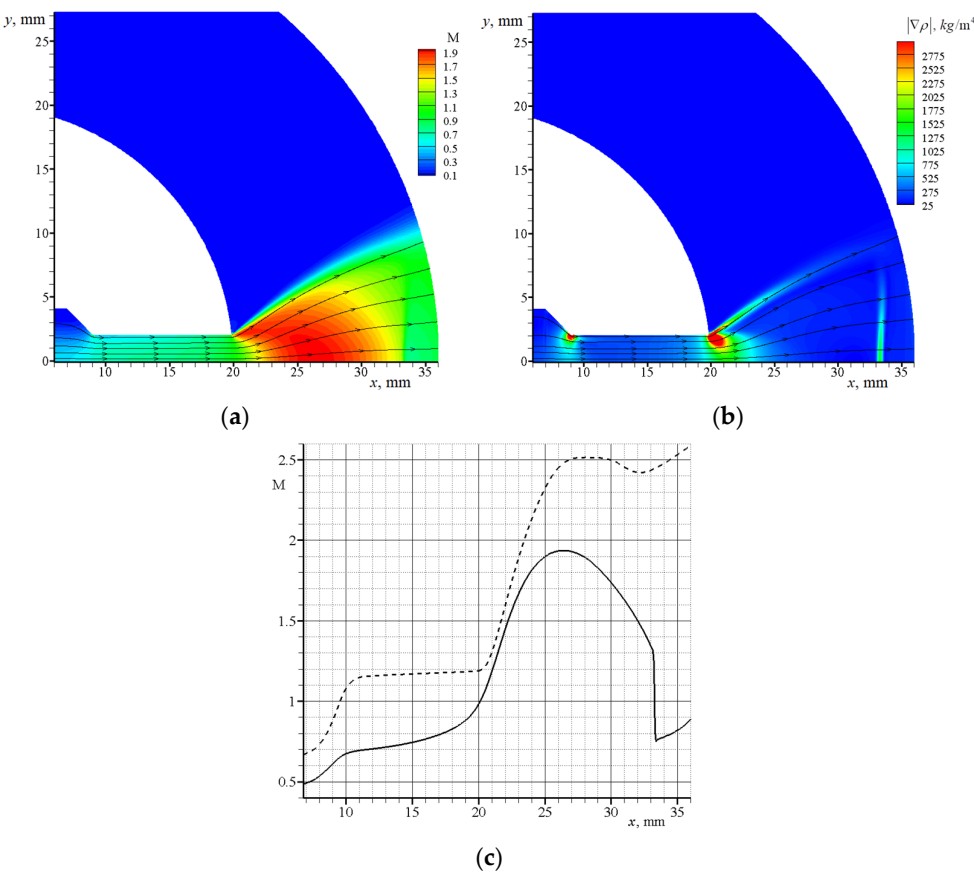

**Figure 5.** Steady-state gas flow in the channel in the case of $h = 0.2$ mm: (**a**) Mach-number isolines and gas streamlines; (**b**) isolines of density gradient $|\nabla \rho|$; (**c**) Mach-number profile on the axis $y = 0$; the dashed line shows data calculated without taking into account the friction force acting on the gas from the side of the disks.

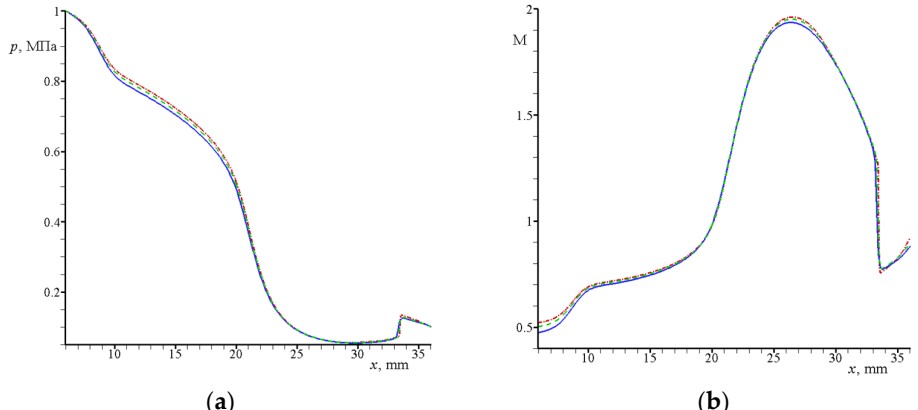

**Figure 6.** Distribution of pressure (**a**) and Mach number (**b**) on the axis $y = 0$ in the steady gas flow calculated with different numbers of cells: the solid, dashed, and dash-and-dot lines correspond to $N_c = 51,783$, $N_c' = 207,132$, and $N_c'' = 828,528$, respectively.

Figure 7 compares the calculation results with the experimental data. To estimate the error in the experiment, several experiments were carried out with various external disks.

Different symbols in Figure 7a (circles and squares) show the results of static-pressure measurements performed on disks that differed in the location of pressure holes.

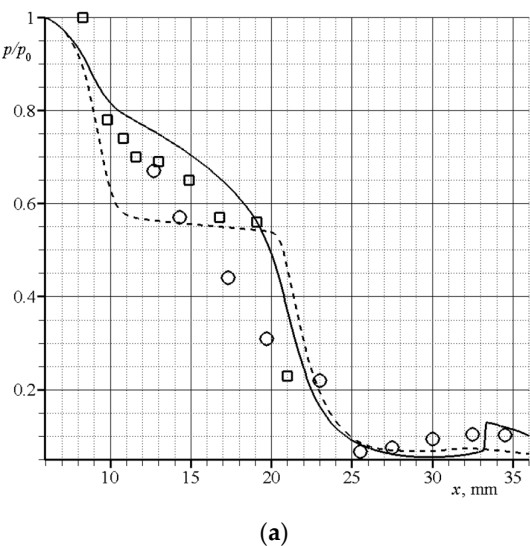

(**a**)

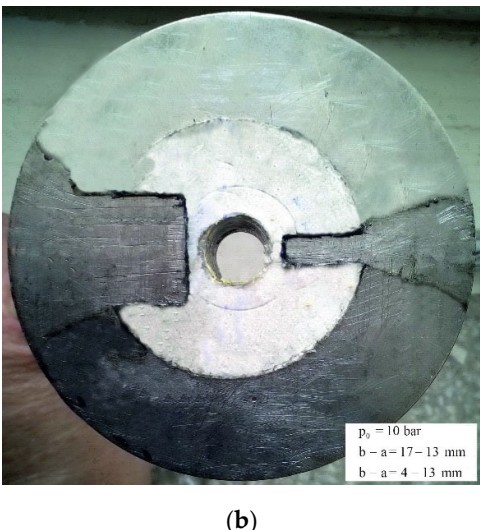

(**b**)

**Figure 7.** Results of experiments and calculations performed at $h = 0.2$ mm: (**a**) distributions of static pressure on the surface of the outer disk on the axis $y = 0$ in the experiment (symbols) and in the calculations (solid line); (**b**) soot-oil coating on the surface of the outer disk in the presence of two channels of width $b = 17$ mm and $b = 4$ mm.

The solid line in Figure 7a shows the calculated distribution of pressure along the jet axis. The dashed line shows the curve calculated without taking into account the friction force due to the disks, $\tau_x = 0$, $\tau_y = 0$. Evidently, within the experimental error, a satisfactory agreement between the calculated and experimental results is observed. Figure 7b shows the soot-oil coating shoot after the experiment on the outer disk without pressure holes yet with two channels. The boundaries of the channel and the jets exhausting into the flooded slot space correspond to the black lines in Figure 7b. This case corresponds to a fan-shaped right jet, whose opening angle at the upper boundary of the jet is 25°, and at the lower boundary −30°. A certain scatter in the angles of the jet opening is apparently due to the non-parallelism of the disks. The calculated jet shown in Figure 5a is also fan-shaped with a half-angle of 30°. Both in calculations and experiments, a supersonic underexpanded jet exhausting into a flooded slot space has a fad-like shape. This was due to the action of the friction force on the jet from the side of the discs. In the absence of disks, the supersonic underexpanded jet exhausting into the flooded space has a barrel shape.

Note that the numerical calculations described above were performed on a personal computer. One calculation took several minutes. The profound gain in calculation time was due to the fact that in our 2D model there was no need to calculate the boundary layer, in which case, in order to fulfill the condition $y_+ \leq 1$, it was necessary to use a refined difference grid. Instead, the 2D Models (3) and (5) use Formula (4) for the drag coefficient $C_W$. The values of the coefficients entering the latter formula are to be borrowed either from the experiment or from a special calculation of viscous gas flow.

### 3.2. Calculated Data for a Gas–Particle Jet

#### 3.2.1. Jet in between Narrow Disks

The ratio of the outer radius $r_e = 36$ mm to the inner radius $r_2 = 20$ mm of the nerrow disks is $r_e/r_2 = 1.8$. Figure 8 shows the pattern of the steady flow in a two-phase gas–particle jet in the case when the width of the channel and slotted space was small, $h = 0.2$ mm. Here, the particle flow rate was $j_s = 0.584 \cdot 10^{-3}$ kg/s.

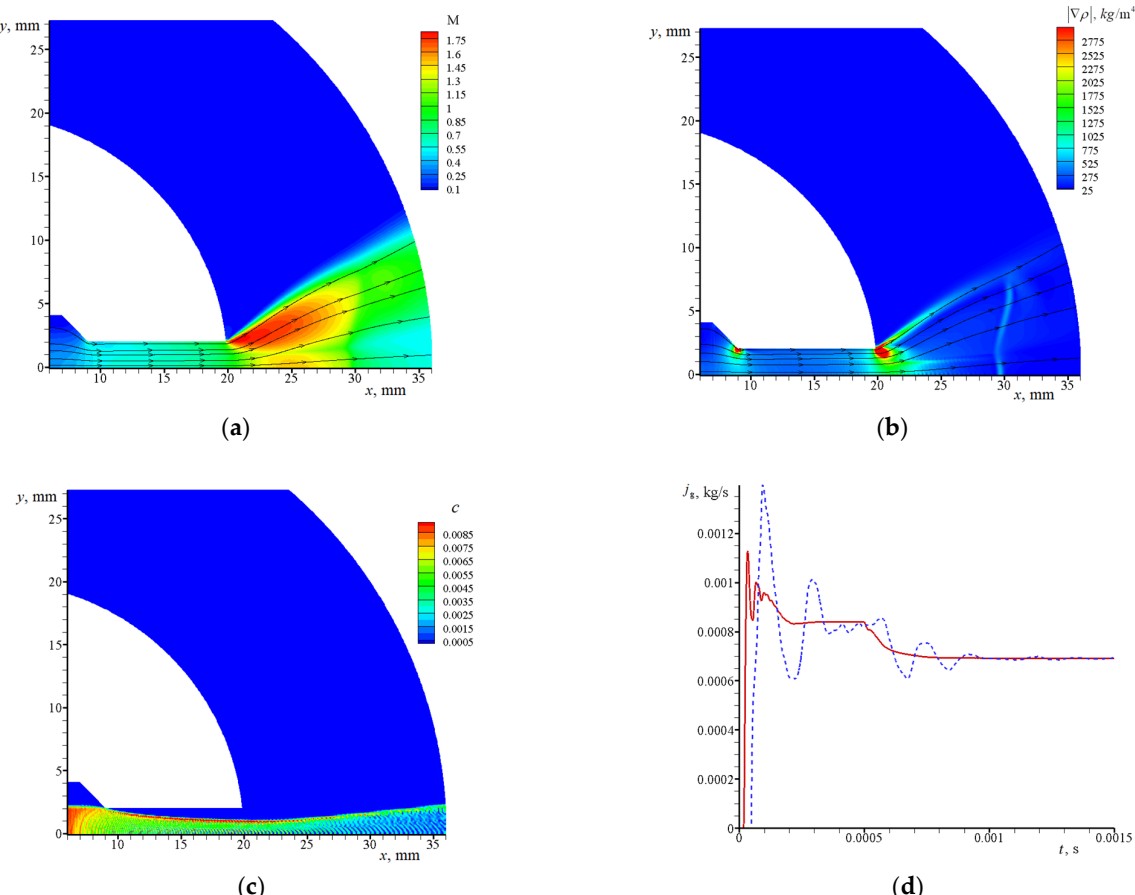

**Figure 8.** Calculated data for a two-phase gas–particle jet exhausting from a channel into a slotted space with $h = 0.2$ mm: (**a**) Mach-number isolines and gas-flow streamlines; (**b**) isolines of density gradient $|\nabla\rho|$; (**c**) particle concentration; (**d**) dependence of the gas flow rate at the channel inlet (solid line) and at the slot-space outlet (dashed line) on time.

Figure 5 shows the established pattern of the gas flow without particles and with the same parameters at the inlet to the channel except for the channel width, which was $h = 0.2$ mm. From Figure 5a,b and Figure 8a,b, it follows that in both cases, when the gas exhausts from the channel, it accelerates to a supersonic velocity in a central expansion wave emerging from the corner point. Particles decelerate the supersonic gas jet and, as a result, the maximum Mach number M = 1.75 in the two-phase gas–particle jet turns out to be lower than that in the gas jet without particles. Due to the friction about the disk walls and gas deceleration by particles, there forms a curvilinear shock wave closing the supersonic flow. Comparison between Figures 5 and 8 shows that the presence of particles in the supersonic jet leads to a displacement of the closing shock wave in the upstream direction, towards the exit cross-section of the channel. Furthermore, the particles present in the jet lead to the formation of a gas-flow non-uniformity in the transverse cross-section (see Figure 8a,b). This non-uniformity is formed during the interaction of the expansion wave with the boundary of the particle jet, and then it becomes more pronounced due to the gas deceleration by particles. In the part of the gas region occupied by particles, a more intense gas deceleration occurs compared with the region free of particles. As a result, the jet splits into an internal jet with particles and an external jet without particles (see Figure 8a,b). The internal and external jets are separated by a tangential discontinuity that coincides with the particle-jet boundary. Note that the criterion for establishment of the flow in the two-phase gas–particle jet was also the stabilization of gas-flow rate at the exit from the slot space with $r_e = 35$ mm (see Figure 8d).

The particle jet in the channel and in the slotted space is shown in Figure 8c. It is seen that in the channel, in the subsonic gas stream, the particle jet undergoes compression. On the contrary, in the slot space this jet undergoes expansion that occurs due to its interaction with the gas jet. The distribution of the concentration of particles in the transverse cross-section of the particle jet is essentially non-uniform. On the upper boundary of the particle jet, there forms a lateral caustic in which the density of particles increases by about two to three times (see Figure 9). As the flow moves downstream, in the supersonic region, the particle jet undergoes expansion in a transverse direction. As a result, the concentration of particles on the caustic decreases in value, and the caustic itself shifts in the direction of the ordinate axis (see Figure 9). In the cross-section $x = 23$ mm, the caustic is at the point with the coordinate $y = 1$ mm, and in the cross-section $x = 30$ mm, at the point $y = 1.6$ mm.

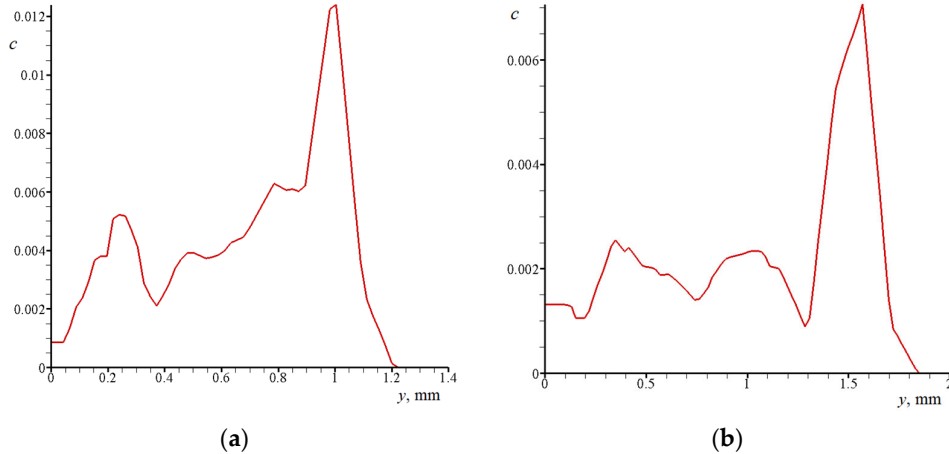

(**a**)  (**b**)

**Figure 9.** The distribution of particle concentration across the flow in two cross-sections: (**a**)—$x = 23$ mm; (**b**)—$x = 30$ mm.

In order to exclude the influence due to the friction force acting from the side of the disks on the two-phase flow, we performed calculations for a large inter-disk spacing, $h = 0.4$ mm, at the rate of particle consumption being $j_s = 1.168 \cdot 10^{-3}$ kg/s. The calculated data for Mach-number isolines are shown in Figure 10.

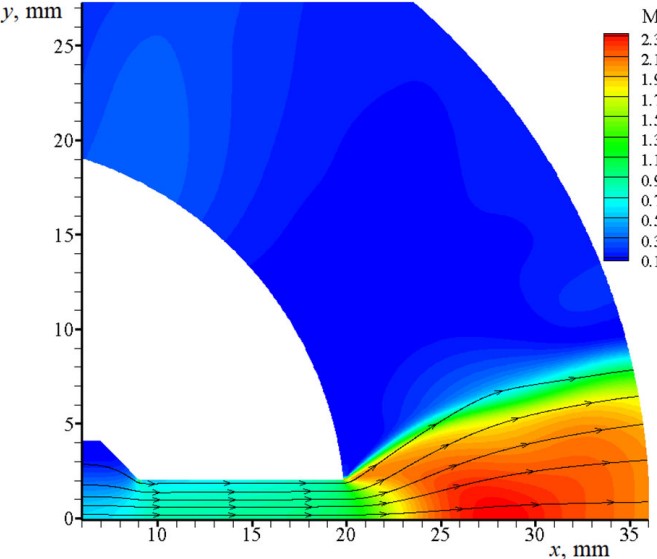

**Figure 10.** Calculated Mach-number isolines for $h = 0.4$ mm.

The comparison of Figure 8a ($h = 0.2$ mm) with Figure 10 ($h = 0.4$ mm) shows that in both cases the qualitative pattern of the flow is the same. However, on increasing the inter-disk spacing, the Mach number in the supersonic jet has increased, and the closing shock wave has shifted downstream. Figure 11 shows the dependences of the gas velocity $u = u(x)$ and the velocity of particles $u_p = u_p(x)$ along the abscissa axis calculated for the cases of $h = 0.2$ mm and $h = 0.4$ mm. Evidently, on increasing the inter-disk spacing, the velocity of the gas flow and that of particles have both increased accordingly.

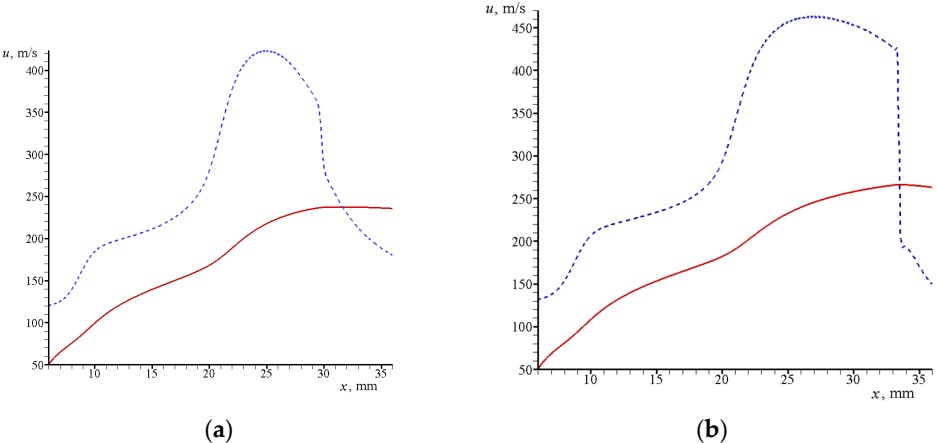

(a)　　　　　　　　　　　　　　　(b)

**Figure 11.** Calculated gas velocities $u = u(x)$ (dashed lines) and particle velocities $u_p = u_p(x)$ (solid lines) at $h = 0.2$ mm (**a**) and $h = 0.4$ mm (**b**).

The effects observed in Figures 10 and 11 are related to the reduction in the friction force acting from the side of the disks on the supersonic jet.

To assess the influence due to the grid step, the same problem was solved for the flow of a gas–particle mixture between the disks. In the numerical calculation, the number of gas cells and the number of particles were increased by 4 times ($N'_c = 207,132$ and $N'_s = 16,584$). Figure 12 compares the data calculated for $N'_c = 207,132$, $N'_s = 16,584$ and $N_c = 51,783$, $N_s = 4146$.

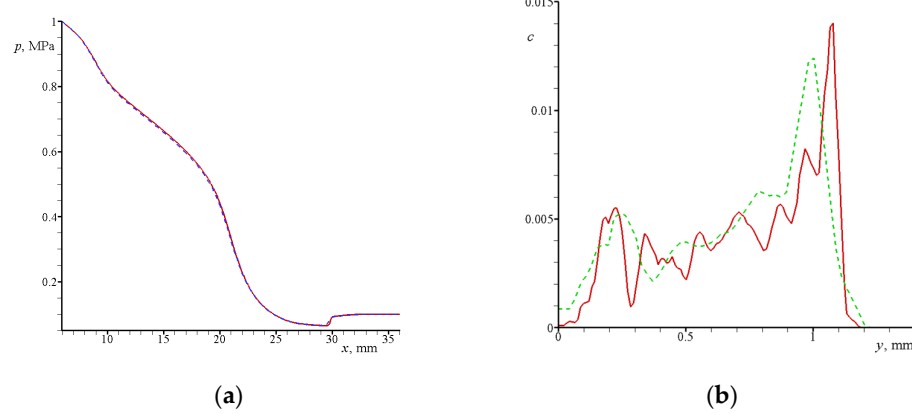

(a)　　　　　　　　　　　　　　　(b)

**Figure 12.** Comparison of data calculated for two values of the number of the gas and particle cells: (**a**) distribution of pressure on the jet axis; (**b**) distribution of particle concentration in the section $x = 23$ mm. Data shown with solid lines were obtained in the calculation with $N'_c = 207,132$ and $N'_c = 16,584$, and data shown with dashed lines, in the calculation with $N_c = 51,783$ and $N_s = 4146$.

From Figure 12, it follows that the distributions of gas pressure in these two calculations coincide with each other. An increase in the number of particle cells led to a more detailed pattern in the distribution of particle concentration $c(y)$ and to a slight shift in the center of the caustic $\Delta x_k / x_k \approx 6\%$.

### 3.2.2. Jet in between Widely Spaced Disks

The ratio of the outer radius $r_e = 90$ mm to the inner radius $r_2 = 20$ mm of the widely spaced disks is $r_e/r_2 = 4.5$. The revealed pattern of the flow in a two-phase gas–particle jet for the case in which the width of the channel and that of the slotted space (separation between the disks) were small, $h = 0.2$ mm, is shown in Figure 13.

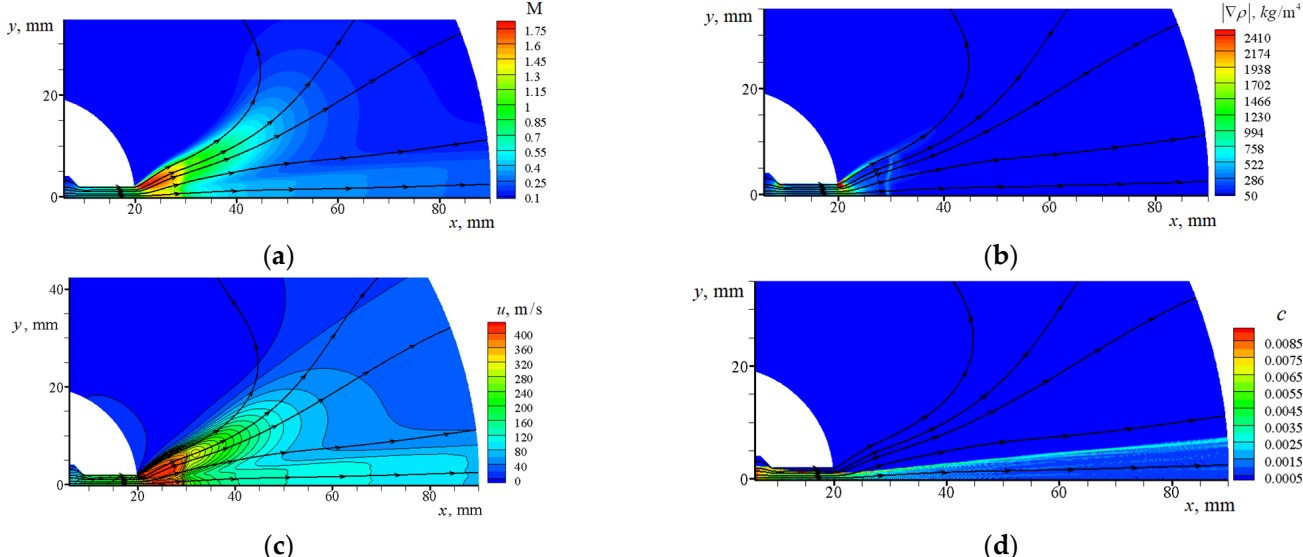

**Figure 13.** Calculated data for a two-phase gas–particle jet exhausting from a channel into a slotted space with $h = 0.2$ mm: (**a**)—Mach-number isolines $M(x,y)$ and gas-flow streamlines; (**b**)—isolines of density gradient $|\nabla\rho|$; (**c**)—longitudinal gas velocity $u(x,y)$ and gas-flow streamlines; (**d**)—concentration of particles $c(x,y)$.

From Figure 13a, it is seen that, as the gas moves in the slot space between widely spaced disks, the presence of particles leads to the splitting of the gas jet into two jets, an internal and external one, similar to the case of the flow in between closely-spaced disks (see Figure 8). The internal jet is located inside the particle jet, and its upper boundary therefore coincides with the boundary of this jet. The external jet is located above the jet of particles. When exhausting out of the channel, both jets, separated by a tangential discontinuity, undergo acceleration to supersonic velocity. Under the action of the force of friction due to the disk walls, a curvilinear shock forms in the cross-section $x_* \approx 30$ mm (see Figure 13b). In the internal jet, the shock is normal, and the flow behind the shock in the internal jet is therefore subsonic. In the external jet, the shock is oblique, and the flow behind this shock is therefore supersonic. Due to the deceleration by the disk walls, the flow in the outer jet becomes subsonic. As the flow moves downstream, the external subsonic jet undergoes expansion, and the gas velocity in this jet quickly decreases (see Figure 13c). As a result, at large distances from the channel exit plane, the velocity in the internal jet turns out to be higher than that in the external jet (see Figure 8c). In the case of widely spaced disks (see Figure 13d), the particle jet has the same structure as in the case of closely spaced disks (see Figures 8c and 9). As the flow moves downstream, the particle jet expands according to the linear law.

Figure 14 shows the established pattern of the flow in the gas jet without particles with the same parameters as those at the inlet to the channel.

In the gas jet without particles (see Figure 14a), the flow first accelerates to a supersonic velocity. Afterwards, due to the interaction with the disk walls, a normal shock forms in the jet, behind which the flow decelerates to a subsonic velocity. From Figures 13 and 14, it is evident that in the case of wide disks $(r_e/r_2 = 4.5)$, an exponential lateral spreading of the jet occurs behind the shock wave. In the case of narrow discs $(r_e/r_2 = 1.8)$, the jet width increases linearly (see the calculated data in Figures 5 and 8 and the experimental

data in Figure 7b). A similar flow pattern is observed in the experiment (see Figure 15) for a gas jet in between wide disks ($r_e/r_2 = 3.8$) under the same conditions in the prechamber ($p_0 = 1$ Mpa , $T_0 = 283$ K).

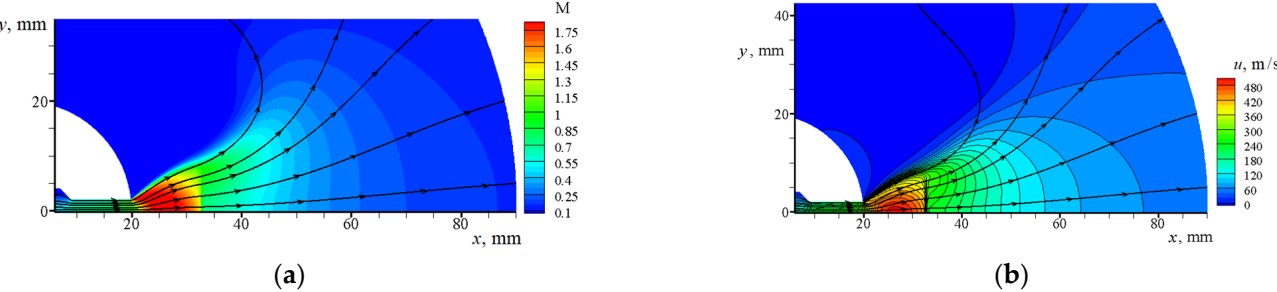

(**a**)　　　　　　　　　　　　　　　　　　(**b**)

**Figure 14.** Calculated data for a gas jet without particles exhausting from a channel into a slot space with $h = 0.2$ mm: (**a**)—Mach-number isolines $M(x,y)$ and gas-flow streamlines; (**b**)—longitudinal gas velocity $u(x,y)$ and gas-flow streamlines.

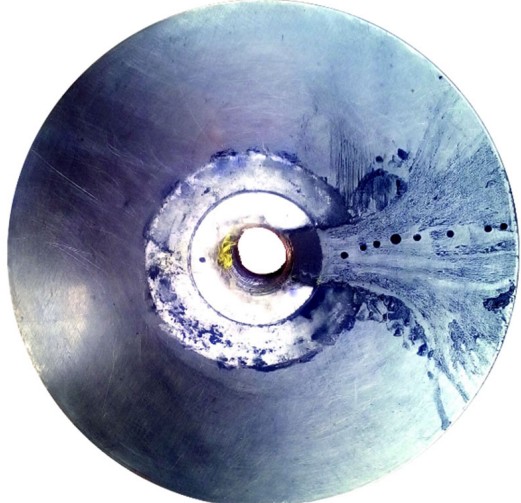

**Figure 15.** Soot-oil coating on the surface of the outer wide disc at $h = 0.15$ mm.

This difference between the flows in the slot space in between the narrow and wide disks is due to the different position of the shock wave in the two cases. In the case of the narrow disks, the shock wave is located at the end of the slot space (see Figures 5 and 8), so that the gas flow between the disks turns out to be supersonic for the most part. The expansion of the jet in this case occurs due to the rotation of the velocity vector in the centered rarefaction wave. In the case of wide disks (see Figures 13 and 14), the shock wave is located at the beginning of the slot space; that is why the gas flow behind the shock wave is subsonic in a larger region of the slot space.

Figure 16 shows the distributions of gas parameters in three cross-sections of the gas jet: in front of the shock wave (solid line) and behind the shock wave (dashed and dash-and-dot lines).

It is seen that the gas parameters depend on the distance to the jet axis $y$. The average values $\bar{p}(x)$, $\bar{\rho}(x)$ and $\bar{u}(x)$ were calculated using the formulas

$$\bar{\rho} = \frac{1}{y_j}\int_0^{y_j} \rho \, dy \, , \ \bar{u} = \frac{1}{\bar{\rho}y_j}\int_0^{y_j} \rho u \, dy \, , \ \bar{p} = \frac{1}{\bar{\rho}y_j}\int_0^{y_j} \rho u \, dy \qquad (9)$$

where $y_j$ is the jet width, which was determined from the condition $u(x, y_j) = \alpha u(x, y = 0)$, $\alpha = 0.3$. Figure 17 shows the distributions of these values

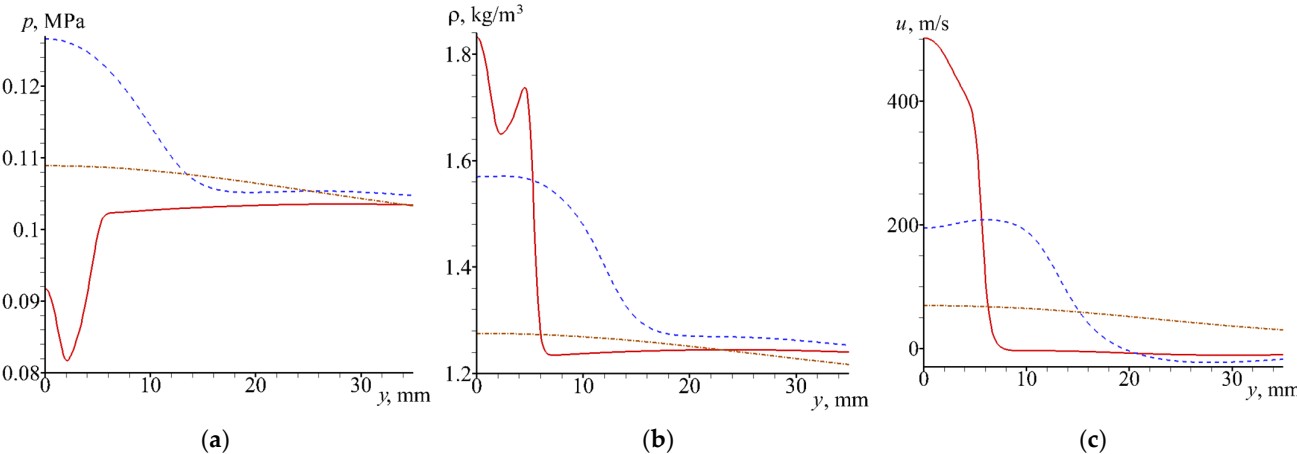

**Figure 16.** The distribution of gas parameters across the jet in the gas flow between wide disks: (**a**)—pressure $p(y)$; (**b**)—density $\rho(y)$; (**c**)—longitudinal gas velocity $u(y)$. The solid red lines correspond to the section $x = 25$ mm; the dashed blue lines, to the section $x = 40$ mm; and the dash-and-dot brown lines, to the section $x = 70$ mm.

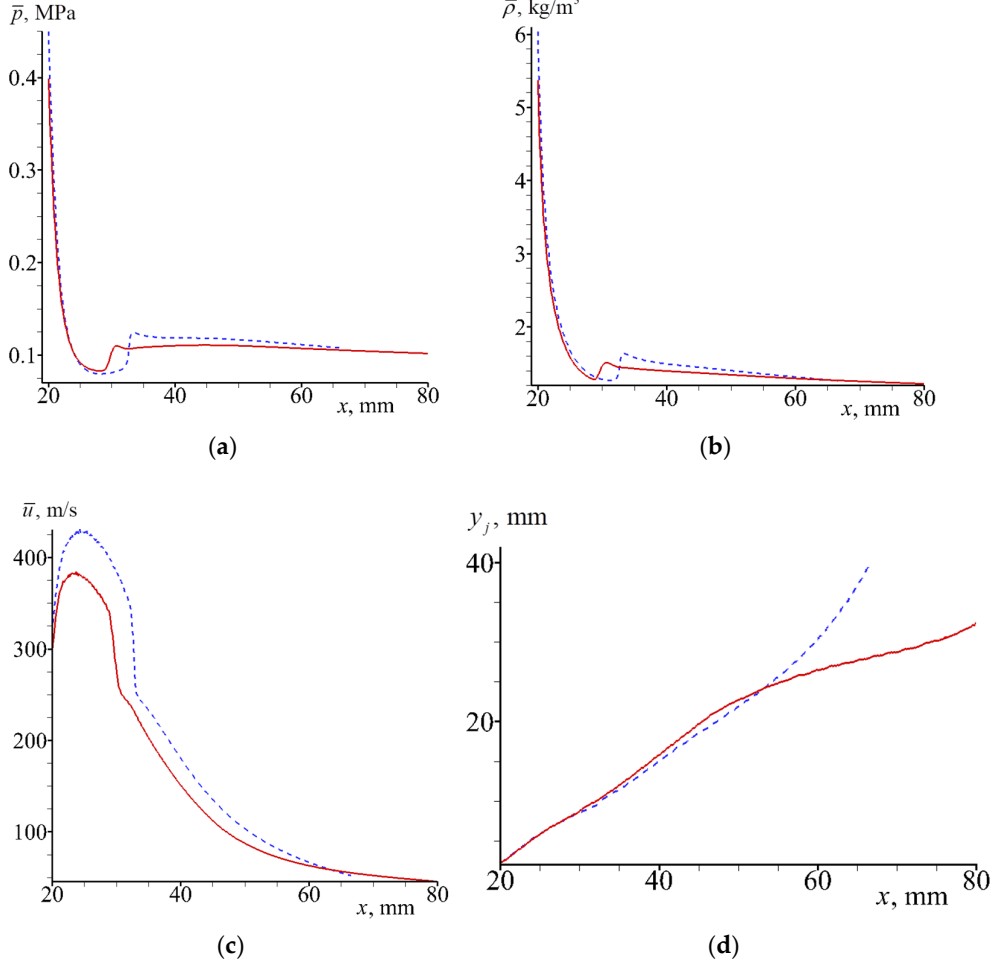

**Figure 17.** The distribution of average gas parameters in the flow between wide disks: (**a**)—$\overline{p}(x)$; (**b**)—$\overline{\rho}(x)$; (**c**)—$\overline{u}(x)$; (**d**)—$y_j(x)$. The solid red lines correspond to a two-phase jet of gas–particle mixture, and the dashed blue lines to a gas jet without particles.

The coordinates of shock waves in the gas jet and the two-phase jet in Figure 17 are close to each other, $x_* \approx 33$ mm. It is seen that behind the shock wave ($x > x_*$), the pressure and density vary slightly ($\partial \overline{p}(x)/\partial x \approx 0$ and $\partial \overline{\rho}(x)/\partial x \approx 0$), and the velocity rapidly decreases with increasing coordinate $x$. The jet width of the gas–particle mixture increases linearly, and the exponential expansion of the stationary gas jet width without particles take place.

Let us give a qualitative explanation as to the exponential expansion of the stationary gas jet without particles in the case of wide disks. It is seen from Figures 13a and 14a that there is a subsonic flow behind the shock wave. To describe the flow in between wide disks, we use the quasi-one-dimensional approximation. Since in the jet we have $u > v$, we neglect the transverse gas velocity in Equation (3). Averaging Equation (3) over the transverse coordinate $y$ and taking into account the fact that $\partial \overline{p}(x)/\partial x \approx 0$, we obtain

$$\overline{\rho u} y_j h = B , \quad \overline{u} \frac{d\overline{u}}{dx} = -C_f \frac{\overline{u}^2}{h} , \quad C_P \overline{T} + \frac{\overline{u}^2}{2} = C_P T_0 \tag{10}$$

where the bar denotes the average gas parameters, $y_j = y_j(x)$ is the transverse coordinate of the jet boundary, $B = $ const is the gas flow rate in the jet, and $T_0$ is the stagnation temperature. Equation (10) are valid in the slot space behind the shock wave $x_* < x < r_e$. The solution of Equation (10) is given by the formulas:

$$y_j = \frac{B}{\overline{\rho u} h} , \quad \overline{u} = \overline{u}_* \exp\left(-C_f \frac{(x - x_*)}{h}\right) , \quad \overline{T} = \overline{T}_* + \frac{\overline{u}_*^2}{2C_P}\left(1 - \exp\left(-2C_f \frac{(x - x_*)}{h}\right)\right) , \tag{11}$$

where the asterisk denotes the parameters of the gas behind the shock wave at $x = x_*$. The gas density can be found from the equation of state $\overline{\rho} \approx \overline{p}/R\overline{T}$. As it is evident from Figure 17b, the density of the gas behind the shock wave can be approximately considered constant; therefore, below, we assume that $\overline{\rho} \approx \overline{\rho}_*$. In this approximation, from the first two equations in (11), we obtain the dependence of the jet velocity and width on the longitudinal coordinate:

$$\frac{\overline{u}}{\overline{u}_*} = \exp\left(-C_f \frac{(x - x_*)}{h}\right) , \quad \frac{y_j}{y_j^*} = \exp\left(C_f \frac{(x - x_*)}{h}\right) , \tag{12}$$

It follows from here that the exponential growth of the jet width is due to the deceleration of the gas by the friction on the disk walls, and due to the pressure's remaining constant when the gas moves downstream in the subsonic jet.

Using Formula (11), we estimate the characteristic length of change in the speed and width of the jet, $\Delta x \approx h/C_f$. Behind the shock wave, we have: $u \approx 200$ m/s (Figure 17c). Taking into account the fact that $h = 0.2$ mm and $\nu \approx 0.15 \cdot 10^{-4}$ m$^2$/s, we find the characteristic Reynolds number: $\mathrm{Re} \approx 2.7 \cdot 10^3$. Substituting this value into Formula (4) and then into Formula (11), we obtain for the characteristic length, a value of $\Delta x \approx 25$ mm. The obtained estimate agrees well with the results of the numerical calculation (see Figure 17c).

Let us estimate the effect of particles on the transverse structure of the jet. A comparison of Figure 13c with Figure 14b shows that in a jet without particles, there is a smoother decrease in velocity $u(y)$ with displacement along the $y$-axis (at $x = $ const) in comparison with a two-phase jet (see Figure 18a).

From Figure 18b, it follows that the momentum flux $J_p = \rho u^2 + p$ in the internal two-phase jet exceeds the same flux in the gas jet without particles.

In order to investigate the effect due to the force of friction acting from the side of the disk walls, we performed calculations for a larger inter-disk separation. Figure 19 shows the established pattern of the flow in a gas jet without particles in that case.

From Figure 19, it is seen that the exhausting jet without particles has a structure typical of the exhaustion of supersonic underexpanded jets [18,19]. Within the calculated region, the supersonic gas jet consists of two barrels separated with a shock wave [24].

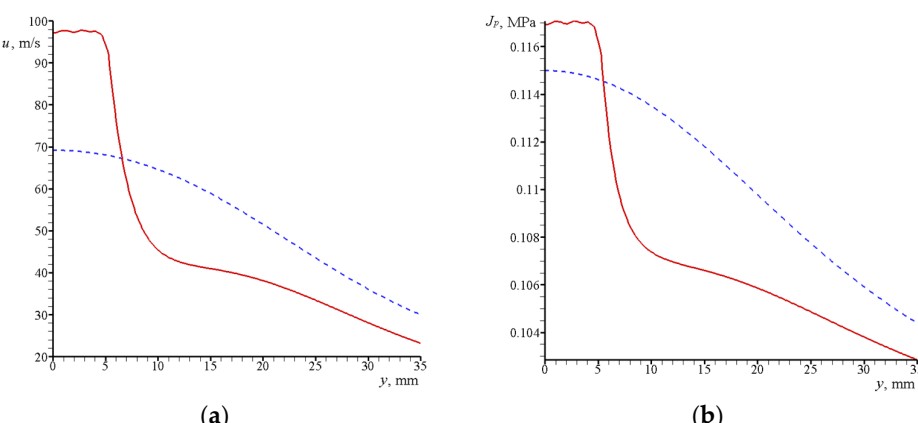

**Figure 18.** The profile of flow velocity $u(y)$ (**a**) and that of momentum flux $J_p(y)$ (**b**) calculated in the cross-section $x = 70$ mm. The solid line and the dashed line show the data, respectively for a two-phase jet and for a gas jet without particles.

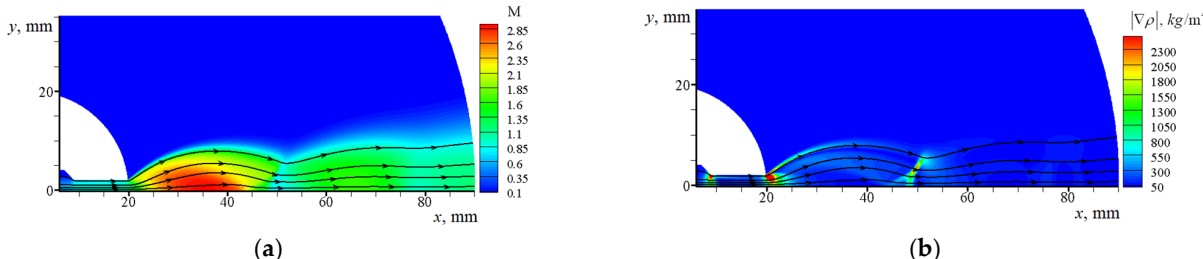

**Figure 19.** Calculated data for a two-phase jet without particles exhausting from a channel into a slotted space with $h = 0.8$ mm: (**a**)—Mach-number isolines $M(x, y)$ and gas-flow streamlines; (**b**)—isolines of density gradient $|\nabla\rho|$.

Figure 20 shows the established pattern of the flow in a two-phase gas–particle jet exhausting into the space with the same separation between the walls, $h = 0.8$ mm.

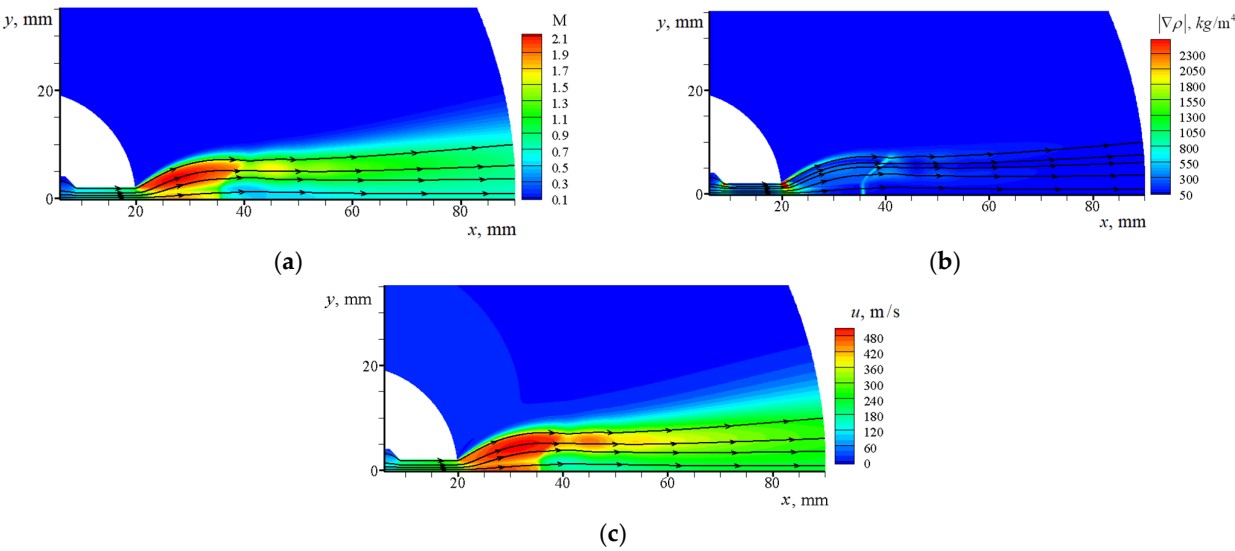

**Figure 20.** Calculated data for a two-phase gas–particle jet exhausting from a channel into a slotted space with $h = 0.8$ mm: (**a**)—Mach-number isolines $M(x, y)$ and gas-flow streamlines; (**b**)—isolines of density gradient $|\nabla\rho|$; (**c**)—longitudinal gas velocity $u(x, y)$ and gas-flow streamlines.

From Figure 20, it is seen that the structure of the two-phase gas–particle jet considerably differs from the structure of the gas jet without particles. The jet exhausting from the channel is accelerated to a supersonic velocity and also splits into two jets. The first (internal) jet is located in the particle region, while the external jet moves in the region over the particle jet. In this case, no barrels that arise when an underexpanded jet exhausts into a submerged space are observed in the gas flow (see Figure 19). At $x \approx 35$ mm, in the flow there arises a curvilinear shock wave, which consists of two characteristic sections. In the region of the internal jet, the shock is normal; therefore, the flow behind this shock is subsonic. In the region of the second (external) jet, the shock wave is oblique, and the flow behind the latter jet therefore remains supersonic. Since, in the latter case, no deceleration of the external jet occurs, the velocity of this jet exceeds the velocity of the internal jet (see Figure 21a). Figure 21 shows the distribution of gas velocity and momentum flux across the flow for a two-phase jet and in a gas jet without particles.

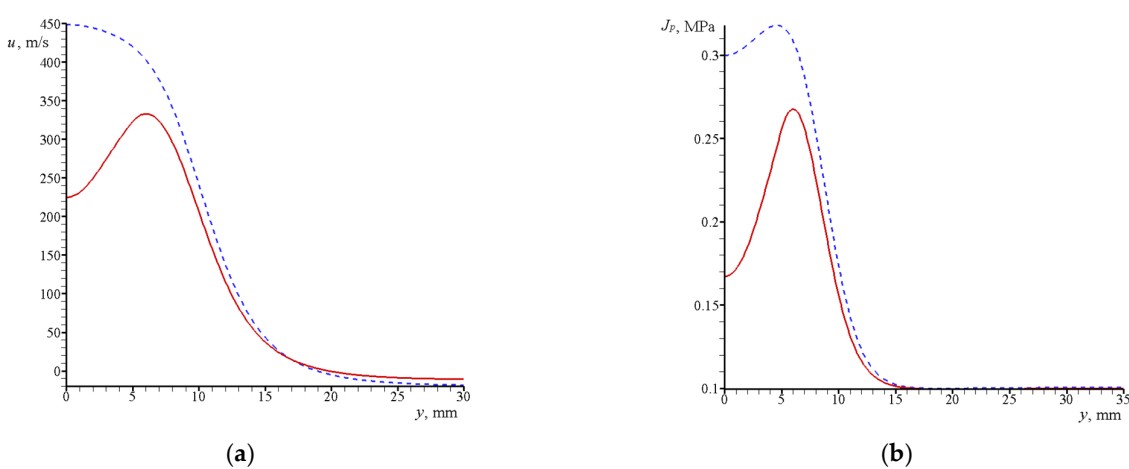

**(a)**                                         **(b)**

**Figure 21.** Dependences calculated in the transverse cross-section $x = 70$ mm: (**a**)—velocity $u(y)$ (**b**)—momentum flux $J_p(y)$. The solid line and the dashed line show the data for a two-phase jet and for a gas jet without particles, respectively.

Figure 21a shows that in the two-phase jet, the gas velocity in the internal jet is two times lower than that in the jet without particles (see Figure 21a). The momentum of the internal jet is also half that in the jet without particles (see Figure 21b).

## 4. Conclusions

The present paper describes a modified Lax–Wendroff scheme making it possible to considerably suppress the oscillations arising at discontinuities. A simplified 2D model for calculating two-phase gas–particle flows in a slotted space has been developed. The model can be used for fast calculation and estimation of supersonic-flow parameters in a slotted space. A numerical simulation of two-phase gas–particle jets exhausting from a channel into a submerged slot space between variously spaced disks is performed. At a small separation between the disks, the friction force acting on the gas from the side of the disk walls exerts a significant influence on the gas flow. At a large disk width, the friction force leads to an exponential growth in the transverse size of the jet when moving downstream.

It is shown that the presence of particles leads to the formation of gas-jet non-uniformity in the transverse cross-section. In the slot space in between the disks, the gas jet separates into two jets, with a tangential discontinuity formed in between. The internal jet is located inside the jet of particles, and the external jet, above this jet. When the flow exhausts out of the channel, the gas undergoes acceleration to supersonic velocity in both jets. The supersonic gas flow ends with a shock, which is a normal one for the internal jet and an oblique one for the external jet. The flows in the two jets differ qualitatively in the cases of small and large inter-disk separations.

A new effect was found in the case of a small inter-disk separation: the velocity of the internal jet turns out to be larger than that of the external jet. This is due to the fact that in the region behind the shock wave, a deceleration of the outer jet and its expansion to the lateral sides occur. As a result, the velocity in the external jet decreases faster than the velocity in the internal jet. In the latter case, the presence of particles leads to the formation of an internal high-velocity two-phase gas–particle jet in the vicinity of the centerline.

In the case of large spacing between the disks, the influence due to the friction force exerted on the disks can be neglected and, as a result, the gas velocity in the internal jet becomes lower than that in the external jet. This is due to the fact that the gas deceleration in the internal jet occurs behind a normal shock wave, whereas that in the outer jet occurs behind an oblique shock wave. In the latter case, the external supersonic jet no longer undergoes deceleration due to the walls, with its velocity exhibiting only minor changes as the flow moves downstream.

The jet of particles experiences considerable deformation in the slot space due to its interaction with the gas stream. The distribution of the concentration of particles in the jet becomes essentially non-uniform. On the upper boundary of the jet, there forms a caustic, on which the concentration of particles turns out to be two to three times higher than that at the center of the jet.

**Author Contributions:** S.K. was leadership and responsible for the written paper, V.K. was responsible for the written paper performed the calculations, V.Z. was responsible for the written paper performed the experiments. All authors have read and agreed to the published version of the manuscript.

**Funding:** The research was carried out within the state assignment of Ministry of Science and Higher Education of the Russian Federation. The APC was funded by journal Aerospace.

**Institutional Review Board Statement:** Not applicable.

**Informed Consent Statement:** Not applicable.

**Data Availability Statement:** Not applicable.

**Conflicts of Interest:** The authors declare no conflict of interest.

## Nomenclature

| | |
|---|---|
| $t$ | Time |
| $x$ | Longitudinal coordinate axis |
| $y$ | Transversal coordinate axis |
| $u$ | Longitudinal gas velocity |
| $v$ | Transversal gas velocity |
| $q$ | Velocity vector modulus |
| $p$ | Gas pressure |
| $\rho$ | Gas density |
| $E$ | Gas specific internal energy |
| $T$ | Gas temperature |
| $a$ | Speed of sound |
| $\mu$ | Dynamic viscosity |
| M | Mach number |
| $\lambda$ | Thermal conductivity |
| $C_v$ | Heat capacity in the pas at constant volume |
| $C_p$ | Heat capacity in the pas at constant pressure |
| $\gamma$ | Poisson's adiabatic exponent |
| Re | Reynolds number |
| Pr | Prandtl number |
| Nu | Nusselt number |
| $\text{Re}_{12}$ | Relative Reynolds number |

| $M_{12}$ | Relative Mach number |
|---|---|
| $f$ | Distribution function of particles |
| $n$ | Concentration of particles |
| $\rho_P$ | Density of the particle |
| $d_p$ | Diameter of the particle |
| $x_P$ | Longitudinal particle coordinate |
| $y_P$ | Transversal particle coordinate |
| $u_P$ | Longitudinal particle velocity |
| $v_p$ | Transversal particle velocity |
| $T_P$ | Particle temperature |
| $a_{px}$ | Longitudinal particle acceleration |
| $a_{py}$ | Transversal particle acceleration |
| $q_P$ | Flux of heat to the particle |
| $C_d$ | Drag coefficient of the particle |
| $C_s$ | Heat capacity of the particle material |
| $C_w$ | Drag coefficient of the friction force acting from the disk |
| $\tau_x$ | Longitudinal component of the friction force acting from the disk |
| $\tau_y$ | Transversal component of the friction force acting from the disk |
| $F_x$ | Longitudinal component of the friction force acting from the particles |
| $F_y$ | Transversal component of the friction force acting from the particles |
| $Q$ | Flux of energy from particles |
| $N_c$ | Number of gas cells |
| $N_s$ | Number of particle cells |
| $y_j$ | Jet width |
| $J_p$ | Momentum flux |
| $\overline{p}$ | Average pressure |
| $\overline{\rho}$ | Average gas density |
| $\overline{u}$ | Average longitudinal gas velocity |

## Appendix A

In matrix form, the system of 2D gas-dynamics equations can be written as

$$\frac{\partial f}{\partial t} + \frac{\partial F}{\partial x} + \frac{\partial G}{\partial y} + H = 0 \tag{A1}$$

System (A1) describes smooth flows in the 2D case. In calculating smooth flows, Equation (A1) should be replaced with an appropriate difference scheme to be solved on a computer. If discontinuities (shock waves or contact discontinuities) occur in the flow, then Equation (A1) at discontinuities becomes inapplicable. Two approaches are then used to calculate such flows. In the first approach, discontinuities in the numerical solution, at which algebraic relations resulting from the laws of conservation of mass, momentum, and energy must be fulfilled, are treated separately. In the second approach, discontinuities in the numerical solution are smeared out due to the artificial, or scheme, viscosity inherent to difference schemes called shock-capturing difference schemes. Shock-capturing difference schemes of first-order accuracy yield monotonic solutions in which discontinuities look smeared over a finite number of cells (Godunov's theorem). However, if the order of accuracy of a scheme is higher than unity, then oscillations seriously deteriorating the solution arise at discontinuities.

To solve this problem, two approaches were used. In the first approach, TVD schemes [25] were proposed, in which, in the vicinity of extrema and discontinuities, a high-order scheme is reconstructed in such a way that it becomes a scheme of first-order accuracy. To this end, nonlinear limiters are used in such a way that when moving onto the next time layer, the total variation in the solution does not increase ($TV(u^{n+1}) \leq TV(u^n)$, where $TV(u) = \sum_k |u_{k+1} - u_k|$). In the second approach, ENO and WENO schemes [26] were proposed, in which a linear combination of difference patterns (or schemes) with coefficients depending on the solution smoothness are used. The coefficients at the difference patterns

are chosen in such a way that the difference patterns containing a discontinuity are made minimal, while the difference patterns in the region of smooth solution, optimal. Both approaches are progressing actively, with an extensive bibliography on them being available in literature (see monograph [27]).

In this paper, with the example of the Lax–Wendroff scheme [28], we propose a simpler method making it possible to suppress oscillations at discontinuities. Namely, we introduce the difference scheme (4), being a linear combination of a first-order scheme with weight $1/(\eta + 1)$ and a second-order Lax–Wendroff scheme with weight $\eta/(\eta + 1)$. In this version of the method, the parameter $\eta$ remains constant throughout the entire flow region. It is obvious that, using smoothness indicators [29], one can make the parameter $\eta$ variable, so that the scheme becomes a scheme of the first order in the region of discontinuity, and a scheme of the second order in the region of smooth flow.

1.     *Numerical calculation scheme.*

Let us construct a numerical scheme for solving the equation

$$\frac{\partial u}{\partial t} + \frac{\partial F(u)}{\partial x} = 0 \tag{A2}$$

We specify integer nodes of a difference grid $t^n, x_k$ using formulas $t^n = t^{n-1} + \tau$ and $x_k = x_{k-1} + h$, and half-integer nodes, using formulas $t^{n+1/2} = t^n + \tau/2$ and $x_{k+1/2} = x_k + h/2$, where $\tau$ is the step over time and $h$ is the step over space. We denote the variable $u$ at integer nodes as $u_k^n$, and the variable at half-integer nodes, as $u_{k+1/2}^{n+1/2}$. Let the values of $u_k^n$ on the $n$-th time layer be known. It is required to find the values of $u_k^{n+1}$ which approximate Equation (A2) on the $(n + 1)$-th time layer. On the intermediate time layer, $t^{n+1/2}$, the values of $u_{k+1/2}^{n+1/2}$ are defined as

$$u_{k+1/2}^{n+1/2} = \frac{1}{2}\left(u_k^n + u_{k+1}^n\right) - \frac{\tau}{2h}\left(F_{k+1}^n - F_k^n\right), \ F_k^n = F(u_k^n) \tag{A3}$$

We determine the values of the variable $u$ on the $(n + 1)$-th time layer $t^{n+1}$ using the formula:

$$u_k^{n+1} = \frac{1}{\eta + 1}\left(\eta u_k^n + \frac{1}{2}\left(u_{k-1/2}^{n+1/2} + u_{k+1/2}^{n+1/2}\right) - \frac{\tau}{h}\left(\eta + \frac{1}{2}\right)\left(F_{k+1/2}^{n+1/2} - F_{k-1/2}^{n+1/2}\right)\right) \tag{A4}$$

where $0 \leq \eta < \infty$. In the case of $\eta = 0$, Formula (A4) yields a first-order accuracy scheme

$$u_k^{n+1} = \frac{1}{2}\left(u_{k-1/2}^{n+1/2} + u_{k+1/2}^{n+1/2}\right) - \frac{\tau}{2h}\left(F_{k+1/2}^{n+1/2} - F_{k-1/2}^{n+1/2}\right) \tag{A5}$$

In the case of $\eta \to \infty$, Formula (A4) yields a second-order accuracy difference scheme

$$u_k^{n+1} = u_k^n - \frac{\tau}{h}\left(F_{k+1/2}^{n+1/2} - F_{k-1/2}^{n+1/2}\right) \tag{A6}$$

which, together with Equation (A3), is called the two-step Lax–Wendroff scheme. Evidently, the difference scheme (A4) presents a linear combination of schemes (A5) and (A6). The parameter $\eta/(\eta + 1)$ determines the weight with which the second-order accuracy scheme enters Equation (A4).

Let us find the order of approximation and derive the stability condition for the schemes (A3) and (A4). We rewrite Equation (A2) as:

$$\frac{\partial u}{\partial t} + c\frac{\partial u}{\partial x} = 0, \ c = \frac{\partial F}{\partial u}$$

Putting $F = cu$ in the difference schemes (A3) and (A4), on the assumption that $c = \text{const}$ we obtain:

$$u_k^{n+1} = \frac{\eta}{\eta+1}\left(u_k^n - \frac{\kappa}{2}\left(u_{k+1}^n - u_{k-1}^n\right) - \frac{\kappa^2}{2}\left(u_{k+1}^n - 2u_k^n + u_{k-1}^n\right)\right) +$$
$$\frac{1}{4(\eta+1)}\left(\left(u_{k+1}^n + 2u_k^n + u_{k-1}^n\right) - 2\kappa\left(u_{k+1}^n - u_{k-1}^n\right) - \kappa^2\left(u_{k+1}^n - 2u_k^n + u_{k-1}^n\right)\right), \tag{A7}$$

where $\kappa = c\tau/h$ is the Courant (CFL) number. Inserting into Formula (A7) the functions $u_k^{n+1} = u(x_k, t^{n+1})$, $u_{k+1}^n = u(x_k + h, t^n)$, and $u_{k-1}^n = u(x_{k-1}, t^n)$ expanded in the vicinity of $u_k^n = u(x_k, t^n)$ in Taylor series, we obtain the first differential approximation:

$$\frac{\partial u}{\partial t} + c\frac{\partial u}{\partial x} = -\left(\frac{\partial^2 u}{\partial t^2} - c^2\frac{\partial^2 u}{\partial x^2}\right)\tau - \frac{c\kappa h}{4(\eta+1)}\frac{\partial^2 u}{\partial x^2} - \frac{c}{6}\frac{\partial^3 u}{\partial x^3}h^2 \tag{A8}$$

Eliminating the time derivatives, from Equation (A8) we finally obtain:

$$\frac{\partial u}{\partial t} + c\frac{\partial u}{\partial x} = \frac{ch}{4(\eta+1)}\left(1 + \frac{1}{\kappa} - \kappa - \kappa^2\right)\frac{\partial^2 u}{\partial x^2} - \frac{c}{6}\frac{\partial^3 u}{\partial x^3}h^2 \tag{A9}$$

The residual on the right side of Equation (A9) is determined by two terms, of which the first term has the order of $u_{xx}h/(\eta+1)$ and the second term, the order of $cu_{xx}h^2$. It is seen that at $\eta = 0$ the difference scheme (A7) is accurate to the first order $O(h)$, and at $\eta \to \infty$, to the second order $O(h^2)$. In the intermediate case, $0 < \eta < \infty$, the error is of order $O(h/(\eta+1))$. By choosing the parameter $\eta$ large enough, one will be able to substantially improve the accuracy of the scheme under consideration. In numerical calculations of two-phase gas–particle flows described in the article, we assumed that $\eta = 9$.

Let us find the condition of stability for the difference scheme (A7). Representing the solution of Equation (A7) as $u_k^n = \lambda^n \exp(i\alpha k)$, we obtain the following expression for the transition coefficient

$$\lambda = \left(1 - i\kappa\sin\alpha + \kappa^2(\cos\alpha - 1)\right)\eta/(\eta+1) +$$
$$+ \left(\cos\alpha + 1 - i2\kappa\sin\alpha + \kappa^2(\cos\alpha - 1)\right)/2(\eta+1). \tag{A10}$$

The necessary condition for the stability of the scheme is the requirement reported in [30], which for Expression (A10) assumes the form:

$$1 - |\lambda|^2 = 2\sin^2(\alpha/2)\left(1 + 2\kappa^2(\eta + 1/2)\right)/(\eta+1) -$$
$$\sin^4(\alpha/2)\left(1 + 2\kappa^2(\eta + 1/2)\right)^2/(\eta+1)^2 - 4\kappa^2\sin^2(\alpha/2) + 4\kappa^2\sin^4(\alpha/2) \geq 0. \tag{A11}$$

In the neighborhood of the point $\kappa = 1$, inequality (A11) simplifies to

$$1 - |\lambda|^2 = 2\sin^2(\alpha/2)\left(1 + 2\eta\sin^2(\alpha/2)\right)\left(1 - \kappa^2\right)/(\eta+1) \geq 0 \tag{A12}$$

Inequality (A12) implies that the difference scheme (A7) is stable for Courant numbers less than unity:

$$\kappa = c\tau/h \leq 1 \tag{A13}$$

The difference scheme (A3), (A4) constructed for Equation (A2) can easily be generalized to the equations of 1D unsteady gas flow:

$$\frac{\partial f}{\partial t} + \frac{\partial F}{\partial x} = 0, \ f = \begin{pmatrix} \rho \\ \rho u \\ \rho(e + u^2/2) \end{pmatrix}, \ F = \begin{pmatrix} \rho u \\ \rho u^2 + p \\ \rho u(e + p/\rho + u^2/2) \end{pmatrix} \tag{A14}$$

where $p = (\gamma - 1)\rho e$, and $\gamma$ is the adiabatic exponent. Applying the substitution $u_k^n \to f_k^n$ in Equations (A3) and (A4), we obtain:

$$f_{k+1/2}^{n+1/2} = \frac{1}{2}\left(f_k^n + f_{k+1}^n\right) - \frac{\tau}{2h}\left(F_{k+1}^n - F_k^n\right),$$

$$f_k^{n+1} = \frac{1}{\eta+1}\left(\eta f_k^n + \frac{1}{2}\left(f_{k-1/2}^{n+1/2} + f_{k+1/2}^{n+1/2}\right) - \frac{\tau}{h}\left(\eta + \frac{1}{2}\right)\left(F_{k+1/2}^{n+1/2} - F_{k-1/2}^{n+1/2}\right)\right).$$

(A15)

As $\eta \to \infty$, the difference scheme (A15) transforms into the Lax–Wendroff scheme. The system of Equation (A14) has three characteristics, $dx/dt = u$ and $dx/dt = u \pm c$, ($c = \sqrt{\gamma p/\rho}$ is the speed of sound), so that the stability condition (A13) for scheme (A15) can be rewritten as:

$$\kappa = \widetilde{c}\tau/h \le 1 \,, \quad \widetilde{c} = \max|u \pm c| \tag{A16}$$

Consider the problem of discontinuity decay in the 1D nonstationary case. Let gas parameters involving a discontinuity at the point $x = 0.6$ m be specified at the initial moment of time $t = 0$ in the region $0 \le x \le 1$ m as follows:

$$\begin{aligned} p &= 0.9 \text{ MPa}, \quad \rho = 1.5 \text{ kg/m}^3 \,, \quad x < 0.6 \text{ m} \,; \\ p &= 0.1 \text{ MPa}, \quad \rho = 1.2 \text{ kg/m}^3 \,, \quad x > 0.6 \text{ m} \,. \end{aligned} \tag{A17}$$

After disintegration of this discontinuity, a rarefaction wave propagates to the left, and a shock wave, to the right. In our calculations, the computational domain was divided into a total of 500 cells ($N = 500$). Figure A1 shows calculated results for the problem of discontinuity decay obtained using three difference schemes: Figure A1a–c the first-order scheme; Figure A1d–f the Lax–Wendroff scheme; Figure A1g–i the proposed scheme (A15) with $\eta = 9$. The dashed lines in Figure A1 show the distributions of gas parameters that were obtained from the exact solution of the discontinuity-decay problem [27]. Evidently, the use of the first-order accuracy scheme leads to a strong smearing of both the shock-wave front and contact discontinuity. The use of the Lax–Wendroff second-order scheme leads to pronounced oscillations in numerical solution arising at the shock-wave front and contact discontinuity. The proposed difference scheme (A15) makes it possible to notably reduce the width of smearing of the shock-wave front and contact discontinuity compared to the first-order scheme, and the amplitude of shock-wave front oscillations compared to the Lax–Wendroff scheme without any loss of accuracy.

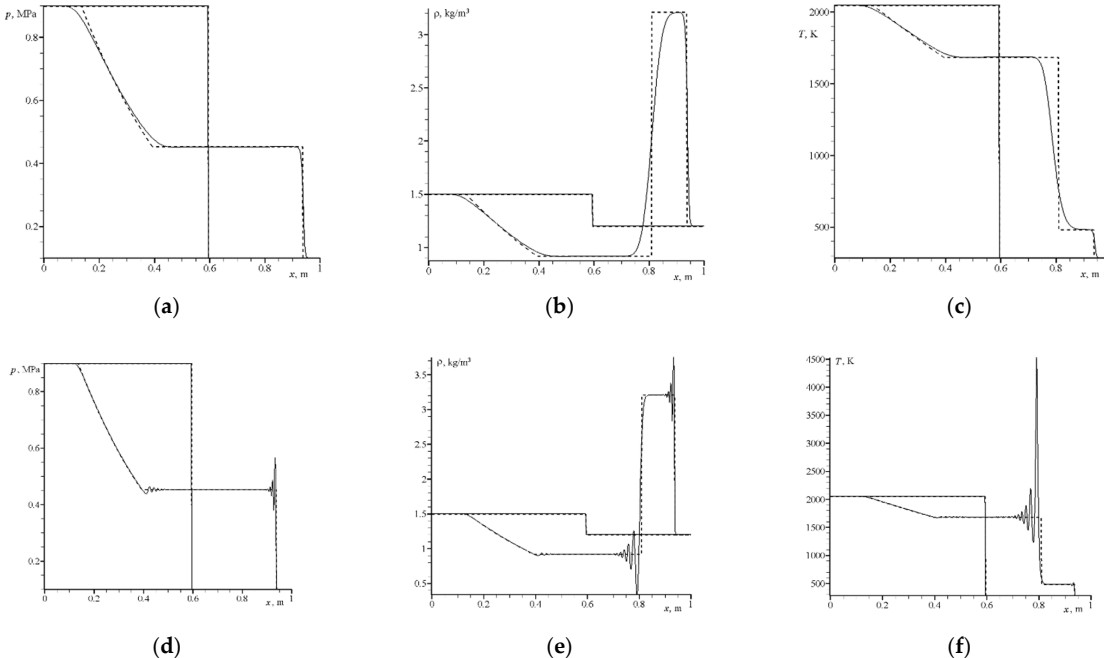

**Figure A1.** *Cont.*

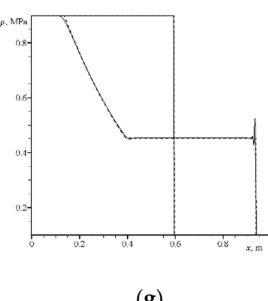
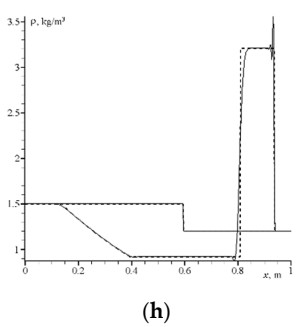
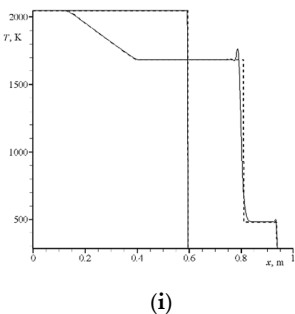

(**g**)          (**h**)          (**i**)

**Figure A1.** Calculation results for discontinuity decay in $p(x)$, $\rho(x)$-, and $T(x)$-distributions at the times $t = 0$ and $t = 5 \cdot 10^{-4}$ s: (**a**–**c**) calculation performed using the first-order scheme; (**d**–**f**) calculation performed using the Lax–Wendroff scheme; (**g**–**i**) calculation performed using scheme (A15) with $\eta = 9$.

2.     *Numerical scheme for 2D calculation on an irregular difference grid.*

When calculating 2D flows in a non-rectangular computational domain, a smooth mapping of this domain onto a rectangle is traditionally performed [29]. In the case of an intricate domain, such a mapping is a difficult task. Consider another approach, in which an irregular grid in the Cartesian coordinate system $(x, y)$ is used to approximate desired functions. Let us generalize the difference scheme (A15) for calculating 2D unsteady flows on an irregular difference grid. We cover the flow region with a grid consisting of quadrangular cells, whose nodes are specified by coordinates $(x_k, y_l)$ (see Figure A2a).

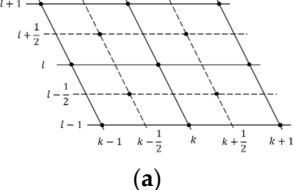
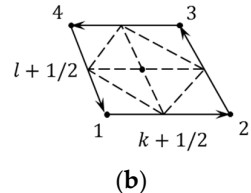
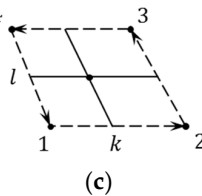

(**a**)          (**b**)          (**c**)

**Figure A2.** Difference grid and cells in a 2D domain: (**a**)—difference grid; (**b**)—cell centered at the point $(x_{k+1/2}, y_{l+1/2})$; (**c**)—cell centered at the point $(x_k, y_l)$.

We choose a cell with coordinates $(x_k, y_l)$, $(x_{k+1}, y_l)$, $(x_{k+1}, y_{l+1})$, and $(x_k, y_{l+1})$, and renumber the nodes of this cell using to the following rule:

$$k, l \to 1 \; ; \; k+1, l \to 2 \; ; \; k+1, l+1 \to 3 \; ; \; k, l+1 \to 4. \tag{A18}$$

In this cell, we construct a parallelogram whose vertices pass through the midpoints of the side faces of the cell. We determine the coordinates of the cell's center at the point $(x_{k+1/2}, y_{l+1/2})$, which coincides with the coordinates of the point of intersection of the diagonals of the parallelogram:

$$\begin{aligned} x_{k+1/2,l+1/2} &= (x_{k,l} + x_{k,l+1} + x_{k+1,l} + x_{k+1,l+1})/4 \, , \\ y_{k+1/2,l+1/2} &= (y_{k,l} + y_{k,l+1} + y_{k+1,l} + y_{k+1,l+1})/4 \, . \end{aligned} \tag{A19}$$

This cell, centered at the point $(x_{k+1/2}, y_{l+1/2})$, is shown in Figure A2b. In the same figure, the parallelogram and its diagonals are shown with dashed lines. The values $f^n_{k+1/2,l+1/2}$ at the center of the cell can be calculated by the formula

$$f^n_{k+1/2,l+1/2} = \frac{1}{4}\left( f^n_{k,l} + f^n_{k+1,l} + f^n_{k+1,l+1} + f^n_{k,l+1} \right) \tag{A20}$$

Similarly, at the center of the cell, we calculate the function $H^n_{k+1/2,l+1/2}$ entering the first equation of (A1). Let flow parameters $f^n_{k,l}$ be known at the nodes of the cells at the

time $t_n$. Using Equation (A1), we modify the difference scheme (A15) to determine the flow parameters $f_{k,l}^{n+1}$ on the next time layer $t^{n+1} = t^n + \tau$. Adding derivatives with respect to one more spatial coordinate, we rewrite scheme (A15) as

$$
\begin{aligned}
f_{k+1/2,l+1/2}^{n+1/2} &= f_{k+1/2,l+1/2}^n - \tfrac{\tau}{2}\left(\left(\tfrac{\partial F}{\partial x}\right)_{k+1/2,l+1/2}^n + \left(\tfrac{\partial G}{\partial y}\right)_{k+1/2,l+1/2}^n + H_{k+1/2,l+1/2}^n\right), \\
f_{k,l}^{n+1} &= \tfrac{1}{\eta+1}\left(\eta f_{k,l}^n + f_{k,l}^{n+1/2} - \tau\left(\left(\eta + \tfrac{1}{2}\right)\left(\left(\tfrac{\partial F}{\partial x}\right)_{k,l}^{n+1/2} + \left(\tfrac{\partial G}{\partial y}\right)_{k,l}^{n+1/2}\right) + \eta H_{k,l}^n + \tfrac{1}{2}H_{k,l}^{n+1/2}\right)\right), \\
f_{k,l}^{n+1/2} &= \tfrac{1}{4}\left(f_{k-1/2,l-1/2}^{n+1/2} + f_{k+1/2,l-1/2}^{n+1/2} + f_{k+1/2,l+1/2}^{n+1/2} + f_{k-1/2,l+1/2}^{n+1/2}\right).
\end{aligned}
\tag{A21}
$$

Using the Stokes formula, we obtain formulas for calculating the derivatives at the centers of the cells (see Figure A2b)

$$
\left(\frac{\partial F}{\partial x}\right)_{k+1/2,l+1/2}^n = \frac{1}{S_{k+1/2,l+1/2}}\oint F\,dy, \quad \left(\frac{\partial G}{\partial y}\right)_{k+1/2,l+1/2}^n = -\frac{1}{S_{k+1/2,l+1/2}}\oint G\,dx,
\tag{A22}
$$

where $S_{k+1/2,l+1/2}$ is the area of the cell with the center at $(k+1/2,\ l+1/2)$, see Figure A2b. For calculating the contour integrals (A22), we use a more convenient enumeration of cell nodes, which is defined above in rule (A18).

In these notations, the cell area $S_{k+1/2,l+1/2} = S_1 + S_2$ is calculated by the formulas

$$
\begin{aligned}
S_1 &= \tfrac{1}{2}\left|\vec{r}_{12}\times\vec{r}_{14}\right| = \tfrac{1}{2}((x_2 - x_1)(y_4 - y_1) - (y_2 - y_1)(x_4 - x_1)), \\
S_2 &= \tfrac{1}{2}\left|\vec{r}_{34}\times\vec{r}_{32}\right| = \tfrac{1}{2}((x_4 - x_3)(y_2 - y_3) - (y_4 - y_3)(x_2 - x_3)).
\end{aligned}
\tag{A23}
$$

The integrals in (A22) are calculated while tracing the boundaries of the cell in a counterclockwise direction:

$$
\begin{aligned}
\oint F\,dy &= F_{12}^n(y_2 - y_1) + F_{23}^n(y_3 - y_2) + F_{34}^n(y_4 - y_3) + F_{41}^n(y_1 - y_4), \\
F_{12}^n &= \tfrac{1}{2}(F_1^n + F_2^n),\ F_{23}^n = \tfrac{1}{2}(F_2^n + F_3^n),\ F_{34}^n = \tfrac{1}{2}(F_3^n + F_4^n),\ F_{41}^n = \tfrac{1}{2}(F_4^n + F_1^n), \\
\oint G\,dx &= G_{12}^n(x_2 - x_1) + G_{23}^n(x_3 - x_2) + G_{34}^n(x_4 - x_3) + G_{41}^n(x_1 - x_4), \\
G_{12}^n &= \tfrac{1}{2}(G_1^n + G_2^n),\ G_{23}^n = \tfrac{1}{2}(G_2^n + G_3^n),\ G_{34}^n = \tfrac{1}{2}(G_3^n + G_4^n), \\
G_{41}^n &= \tfrac{1}{2}(G_4^n + G_1^n).
\end{aligned}
\tag{A24}
$$

The derivatives at the cell nodes $k,\ l$ are calculated similarly. To calculate these derivatives, we surround each cell node $k,\ l$ with a quadrilateral whose nodes are located at the centers of the neighboring cells, $(x_{k-1/2}, y_{l-1/2})$, $(x_{k+1/2}, y_{l-1/2})$, $(x_{k+1/2}, y_{l+1/2})$, and $(x_{k-1/2}, y_{l+1/2})$ (see Figure A2a). Let us renumber the obtained cell nodes using the rule

$$
k - \frac{1}{2}, l - \frac{1}{2} \to 1; k + \frac{1}{2}, l - \frac{1}{2} \to 2; k + \frac{1}{2}, l + \frac{1}{2} \to 3; k - \frac{1}{2}, l + \frac{1}{2} \to 4.
\tag{A25}
$$

This cell centered at the point $(x_k, y_l)$ is shown in Figure A2a. The partial derivatives at the cell node $k,\ l$

$$
\left(\frac{\partial F}{\partial x}\right)_{k,l}^{n+1/2} = \frac{1}{S_{k,l}}\oint F\,dy, \quad \left(\frac{\partial G}{\partial y}\right)_{k,l}^{n+1/2} = -\frac{1}{S_{k,l}}\oint G\,dx
\tag{A26}
$$

can be found using the contour integrals over the cell boundaries calculated by Formulas (A23) and (A24). By direct verification, one can make sure that Formulas (A23)–(A26) define derivatives (A22) with the second-order accuracy.

In the case under study, the computational domain consists of a channel and a slot space bounded by two disks. Due to symmetry, it suffices to calculate the flow in the upper right quarter of the slot space only. Figure A3 shows the grid that covers the flow region in the channel and that in part of the slot space.

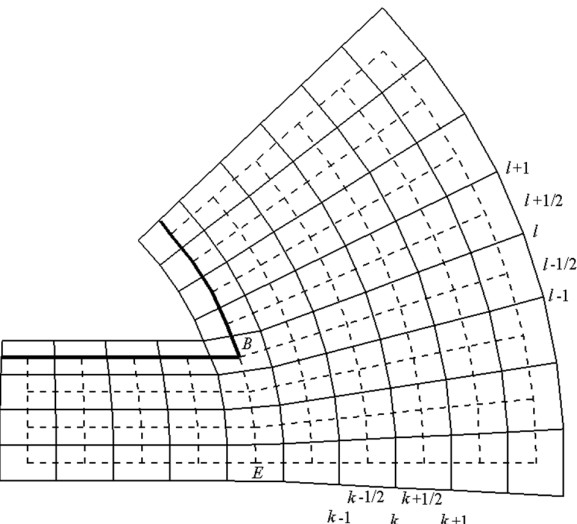

**Figure A3.** Grid in the channel and in the slot space.

The intersections of the dashed lines are the centers of the cells, which are indicated with half-integer indices. The thick lines show the boundaries of the channel and those of the slot space, behind which lawn points (points lying outside the computational domain) are located.

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
