# Peer review of "Numerical Simulation of the Flow in Two-Phase Supersonic Underexpanded Gas–Particle Jets Exhausting into a Slotted Submerged Space"

_aerospace, doi:10.3390/aerospace9080432_

Round 1

Reviewer 1 Report

This is a solid piece of work. The application is very specific, which will limit the impact of the work. However, lessons learned from particle splitting in supersonic jets might find application in other areas (such a rocket propulsion). This reviewer is familiar with the standards of these authors, which is usually very good. This paper is also solid scientifically, I recommend it for publication. 

As feedback, I would like to have seen more quantitative analysis. One could for example define a spread rate of the supersonic jet, location of shocks and spread rate of the particle jet. It would make this paper stronger if some integrated variable were compared for the various cases.

Author Response

Dear Reviewer,

Trying to answer your comments, we have considered the quasi-one-dimensional model of the gas flow between disks for average parameters, namely, for the velocity, density and width of the jet. We obtained a system of equations (10) whose solution is given by formulas (12). As a result, we have shown that the exponential growth of jet width, which was observed (both in calculations and in the experiment) in the flow between wide disks, was due to the deceleration of the gas by the friction force on the disk walls and due to the constant value of pressure observed when displacing downstream in the subsonic jet arising behind the shock wave.

In accordance with comments 2 and 3 of your report, we have considerably expanded the methodological part of our article. Appendix was added to describe the adopted modification of the Lax-Wendroff scheme. Section 2.2, Equations for gas flow in the channel approximation, which describes the 2D model used, was additionally written. Section 3.1, Calculation of the gas flow without particles, presenting a verification of the 2D model, was also added to the article. We hope that the additions made have improved the article and have made it more friendly to the reader.

The authors cordially thank the reviewer for a supporting review of our manuscript.

Sincerely,

Sergei Kiselev

Reviewer 2 Report

The paper presents the results of a numerical simulation of the flow in two-phase gas-particle supersonic jets exhausting into a slotted submerged space formed by two parallel disks. The topic is quite interesting to the reader. The model simulation of a two-phase supersonic in this manuscript version is explained. The simulation results have been discussed numerically and physically. Overall, the manuscript is worthy of publication. However, some crucial issues should be explained further. Therefore, I recommend the paper can be accepted after the following comments are considered in a revised version.

1.      The novelty should be presented. Is it a new numerical method or a new CFD code? I would strongly advise the author to rewrite their introduction by identifying the previous research gap, especially in the numerical simulation of the flow in two-phase gas-particle supersonic jets exhausting topic. Therefore the novelty of the paper can be highlighted.

2.      The simulation results show that the flow structure of the exhausting off-design jet depends on the distance between the disks,  Δz = h, due to the friction force exerted on the disks. However, the computational domain and the presented equation (1,2,3,4) only depend on the x and y-axis. Please explain whether the CFD model is a 2D or 3D Model. Moreover, it is better to present the figure of the meshing model and the boundary condition.

3.      The paper is about CFD Simulation. It is better to present the Grid Independence Test.

4.      The validation of simulation results needs to be presented. It can be compared with experiment results or the other CFD model from previous studies.

5.        The results and discussion should describe the research's major findings and compare them to previous studies. In addition, the current research area's gaps must be highlighted, and future study directions must be presented.

Author Response

  1. Remark: The novelty should be presented. Is it a new numerical method or a new CFD code? I would strongly advise the author to rewrite their introduction by identifying the previous research gap, especially in the numerical simulation of the flow in two-phase gas-particle supersonic jets exhausting topic. Therefore the novelty of the paper can be highlighted.
  2. Answer: In the revised version of our article, an introduction was added to the text to help the reader to better understand the place of our work among other related studies.

Appendix presenting the modification of the Lax-Wendroff scheme in 1D and 2D cases was written. Since the authors never encountered such a modification of this scheme in literature, in the present article they gave its detailed description.

A new section, “2.2. Equations for gas flow in the channel approximation”, was written. In this section, the method of averaging equations over the slot-space thickness is used to derive equations of the 2D model for calculating the gas flow between closely spaced disks. This model is then used in Section 2.3 to describe the two-phase model of the gas-particle flow. Note that this approach was used previously by the present authors in 1D case [2]. In [2], it was shown that the results of calculations of the gas flow between disks performed using the 1D model, are in satisfactory agreement with the calculations performed using the SST k-omega model of turbulence. In the present study, this approach was generalized to the 2D case. We believe that the 2D model proposed in our article will prove useful for researchers of two-phase flows in slot configurations since it allows rather accurate, fast calculations of such flows under limited computing resources.

  1. Remark: The simulation results show that the flow structure of the exhausting off-design jet depends on the distance between the disks, , due to the friction force exerted on the disks. However, the computational domain and the presented equation (1,2,3,4) only depend on the x and y-axis. Please explain whether the CFD model is a 2D or 3D Model. Moreover, it is better to present the figure of the meshing model and the boundary conditions.
  2. Answer: In the present work, 2D approximation was used. The viscous-gas equations were averaged over the width of the slot space. As a result of averaging, equations (3) were obtained, in which viscosity enters the problem through the resistance force at the boundary. The latter force is inversely proportional to the slot width (the distance between the disks). In addition, the slot width is contained in the Reynolds number, the drag coefficient (4) depends on. In fact, Euler equations with the right-hand sides are solved in the -plane. For this reason, in this 2D approximation it is unnecessary to refine the calculation grid in the vicinity of the boundary.

The grid structure is shown in Fig. 3.

  1. Remark: The paper is about CFD Simulation. It is better to present the Grid Independent Test.
  2. Answer: Figures 6, 12 illustrates the independence of calculated data of the mesh-size refinement by 4 and 8 times.
  3. Remark: The validation of simulation results needs to be presented. It can be compared with experiment results or the other CFD model from previous studies.
  4. Answer: The revised version of the article was written together with experimenter V. Zaikovskii. A new section, “3.1. Calculation of gas flow without particles”, was added to the article in which experimental results were presented, see Fig. 7. Figure 7 shows that there exists a satisfactory agreement between the calculated and experimental data.
  5. Remark: The results and discussion should describe the research`s major findings and compare them to previous studies. In addition, the current research area`s gaps must be highlighted, and future study directions must be presented.
  6. Answer: The present work is a development of work [1]. The paper [1] shows the possibility of using radial nozzles for spraying coatings onto inner pipe surfaces. The radial nozzle is formed by two parallel discs, in between which the gas is accelerated together with microparticles. When microparticles impinge onto the pipe surface, particles adhere to it providing that their velocity is high enough. In the present work, the influence of the disk radius and inter-disk spacing on the flow of the two-phase gas-particle jet was studied. Two new interesting effects were discovered for wide disks the distance between which was comparable with, or less than, two tenths of micrometer. First, it is shown (see Figs. 13 and 14) that due to the interaction with particles, the gas jet breaks into two jets. In this case, the speed of the internal gas jet is greater than the speed of the gas jet without particles (it would seem that the particles should slow down the gas, but in reality the opposite is true!!!). Second, both in the calculations (see Figs. 13 and 14) and in the experiment (see Fig. 15), it is shown that the underexpanded gas jet expands exponentially in the case of wide disks (normally, the underexpanded jets are barrel-shaped). An explanation to this phenomenon is given, see solution (12). The results obtained can prove useful in various engineering applications, for example, in spraying and accelerating not only particles, but also liquid droplets.

The authors thank the reviewer for his interesting review that encouraged us to revise our manuscript.      

Reviewer 3 Report

The authors investigate the supersonic flow with particles exhausting a nozzle. Major darwbacks of the manuscript are the lack of motivation description, method description, verification, and validation.

----------

What application are the authors investigating?

What is the point of the "slot"? The outlet boundary conditions are essentially described as free stream conditions. So, a different slot size shold have no impact on the results.

Please provide the characteristics of the mesh. What is the structure? What are the y+ values?

There is no validation or verification of the numerical approach. Please provide a grid sensitifity study.

There is no statement about a turbulence model or why laminar flow is considered. When I calculate the Reynolds number, the jet is well turbulent. Please provide the complete information.

There are many informations missing in the description of the numerical approch. The equations shown are the unsteady Navier-Stokes equations. However, steady-state results are presented. How are they obtained? What was the sampling time?

Several figures show streamlines, where the flow turns around in free space, e.g. figure 8 and 9. How is the physical?

6 out of 19 references are self citations.

There are very few references. No citation to an "aerospace" journal.

* Wang et al. (2021) "Numerical Simulation of Non-Spherical Submicron Particle Acceleration and Focusing in a Converging–Diverging Micronozzle"

* Semlitsch and Mihaescu (2021) "Evaluation of Injection Strategies in Supersonic Nozzle Flow"

* Horner et al. (2021) "Numerical Investigation of a Rectangular Jet Exhausting over a Flat Plate with Periodic Surface Deformations at the Trailing Edge"

Author Response

  1. Remark: What application are the authors investigation?
  2. Answer: This study is important for the Cold Spray technology. Under the action of a supersonic gas flow, particles contained in it are accelerated in a narrow slot space. As a result, a narrow jet of gas-particle mixture is formed at the exit from the slot space. Particles move at a speed of several hundred meters per second. When particles impinge onto an obstacle, a strong narrow coating forms on the obstacle. In the literature, this coating application method is known as the Cold Spray technique [1]. As noted above, the use of round discs makes it possible to adapt this method for applying coating onto inner pipe surfaces [1].
  3. Remark: What is the point of the “slot”? The outlet boundary conditions are essentially described as free stream conditions. So, a different slot size should have no impact on the results.
  4. Answer: The right-hand sides of gas-motion equations (3), (5) include the resistance forces with which the disks act on the gas. The drag forces are inversely proportional to the slot width (the distance between the disks). In addition, the slot width enters the Reynolds number, the drag coefficient (4) depends on.
  5. Remark: Please provide the characteristics of the mesh. What is the structure? What are the values. .
  6. Answer: In this work, 2D approximation was used. The viscous-gas equations were averaged over the slot-space width; as a result of averaging, the viscosity enters Eqs. (3) only through the resistance force at the boundary. In fact, Euler equations with right-hand sides are solved. For this reason, in the present 2D calculations there is no need to refine the mesh near the boundary and calculate . The grid structure is shown in Fig. 3.
  7. Remark: There is no validation or verification of the numerical approach. Please provide a grid sensitivity study.
  8. Answer: Figure 7 shows that there exists a satisfactory agreement between the calculated and experimental data. Figure 6 shows that the calculation results remained unchanged when the mesh was refined by 4 and 8 times.
  9. Remark: There is no statement about a turbulence model or why laminar flow is considered. When I calculate the Reynolds number, the jet is well turbulent. Please provide the complete information.
  10. Answer: As noted above (see the answer to remark 3), Euler equations with right-hand sides were solved, and no turbulence model was used. At the same time, the drag coefficient depends on the Reynolds number, see formula (4). Like in the laminar flows in pipes, at low Reynolds numbers the drag coefficient is inversely proportional to the Reynolds number. Like in the turbulent flows in pipes, at high Reynolds numbers the resistance is inversely proportional to the Reynolds number raised to the power of ¼ (Boussinesq formula). In our article, a linear combination of these two terms was used. The values of the coefficients were chosen from the condition of agreement with the experiment. We draw the reviewer’s attention to our work [2] in which (see Fig. 18 in [2]) a comparison of our calculations of the flow between disks performed using a 2D model (in which the drag force was calculated using formula (4)), with the calculations performed using the SST k-omega model of turbulence was reported. Satisfactory agreement between these calculations was observed. The 2D model proposed in the present article is useful since it allows performing rough yet quick calculations with reasonable accuracy under limited computing resources.
  11. Remark: There are many informations missing in the description of the numerical approach. The equations shown are the unsteady Navier-Stokes equations. However, steady-state results are presented. How are obtained? What was the sampling time?
  12. Answer: Answer: The steady-state flow was calculated using the relaxation method, see Fig. 4a. The numerical calculation was carried out according to the modified Lax-Wendroff difference scheme described in Appendix. The criterion for termination of the calculation process was the reaching of the gas flow rate at which the fluctuations of this rate did not exceed (see Fig. 4b).
  13. Remark: Several figures show streamlines, where the flow turns around in free space, e.g. figure 8 and 9. How is the physical?
  14. Answer: A new effect, the exponential expansion of the jet in the submerged slot space, was discovered in the calculations, - see Figs. 13 and 14 (or Figs. 8 and 9 in the first version of the article). This effect was verified experimentally, see Fig. 15. The paper gives an explanation to this effect. We show that the exponential growth of the jet width, which is observed in the flow between wide disks, is due to the deceleration of the gas caused by the friction against the disk walls and due to the constant value of pressure observed when moving downstream in the subsonic jet arising behind the shock wave.
  15. Remark: There are very few references. No citation to an “Aerospace” journal: Wang et al (2021); Semilitsch and Mihaescu (2021); Horner et al. (2021).
  16. Answer: The above references were added to the revised manuscript.

The authors thank the reviewer for an interesting report.

Round 2

Reviewer 3 Report

The authors addressed my questions and comments.